# An Artificial Intelligence-guided signature reveals the shared host immune response in MIS-C and Kawasaki disease

Pradipta Ghosh [1,2,11✉], Gajanan D. Katkar[1,11], Chisato Shimizu [3,4,11], Jihoon Kim[5], Soni Khandelwal[6], Adriana H. Tremoulet[3,4], John T. Kanegaye [3,4], Pediatric Emergency Medicine Kawasaki Disease Research Group*, Joseph Bocchini[7], Soumita Das[8], Jane C. Burns [3,4✉] & Debashis Sahoo [3,6✉]

Multisystem inflammatory syndrome in children (MIS-C) is an illness that emerged amidst the COVID-19 pandemic but shares many clinical features with the pre-pandemic syndrome of Kawasaki disease (KD). Here we compare the two syndromes using a computational toolbox of two gene signatures that were developed in the context of SARS-CoV-2 infection, i.e., the viral pandemic (ViP) and severe-ViP signatures and a 13-transcript signature previously demonstrated to be diagnostic for KD, and validated our findings in whole blood RNA sequences, serum cytokines, and formalin fixed heart tissues. Results show that KD and MIS-C are on the same continuum of the host immune response as COVID-19. Both the pediatric syndromes converge upon an *IL15/IL15RA*-centric cytokine storm, suggestive of shared proximal pathways of immunopathogenesis; however, they diverge in other laboratory parameters and cardiac phenotypes. The ViP signatures reveal unique targetable cytokine pathways in MIS-C, place MIS-C farther along in the spectrum in severity compared to KD and pinpoint key clinical (reduced cardiac function) and laboratory (thrombocytopenia and eosinopenia) parameters that can be useful to monitor severity.

[1] Department of Cellular and Molecular Medicine, University of California San Diego, San Diego, USA. [2] Department of Medicine, University of California San Diego, San Diego, USA. [3] Department of Pediatrics, University of California San Diego, San Diego, USA. [4] Rady Children's Hospital-San Diego, San Diego, CA, USA. [5] Department of Biomedical informatics, University of California San Diego, San Diego, USA. [6] Department of Computer Science and Engineering, Jacob's School of Engineering, University of California San Diego, San Diego, USA. [7] Willis-Knighton Health System, Shreveport, LA, USA. [8] Department of Pathology, University of California, San Diego, USA. [11]These authors contributed equally: Pradipta Ghosh, Gajanan D. Katkar, Chisato Shimizu. *A list of authors and their affiliations appears at the end of the paper. ✉email: prghosh@ucsd.edu; jcburns@health.ucsd.edu; dsahoo@ucsd.edu

Multisystem inflammatory syndrome in children[1] (MIS-C; initially named Pediatric Inflammatory Multisystem Syndrome Temporally associated with SARS-CoV-2, PIMS-TS)[2] is a rare but severe condition that occurs in children and adolescents ~4–6 weeks after exposure to SARS-CoV-2. First reported in April 2020 in a cluster of children in the United Kingdom[3], followed by other regions of the world[4], the syndrome is characterized by fever, and variably accompanied by rash, conjunctival injection, gastrointestinal symptoms, shock, and elevated markers of inflammation and antibodies to SARS-CoV-2 in the majority of patients. Myocardial dysfunction and coronary arterial dilation may resemble those seen in another uncommon childhood condition, Kawasaki Disease (KD). KD is an acute inflammatory disorder predominantly seen in young children. Since it was first described in Japan[5] in 1967, KD has emerged as the most common cause of pediatric acquired heart disease in the developed world[6]. Little is known about the definitive triggers of KD; what is most widely accepted is that KD is largely an immune response to a plethora of infectious or environmental stimuli including viruses, fungi (e.g., *Candida* sp.), and bacteria[7–9]. The host genetic background appears to shape this idiosyncratic inflammatory response to an environmental antigen exposure[9].

On May 14, 2020, the CDC published an online Health Advisory that christened this condition as Multisystem Inflammatory Syndrome in Children (MIS-C) and outlined a case definition[10]. Since then, as the COVID-19 pandemic spread across many countries, cases of MIS-C soared, with features of shock and cardiac involvement requiring ionotropic support [in the critical care setting]. But distinguishing MIS-C from KD, KD shock syndrome[11], and other severe infectious or inflammatory conditions remains a challenge. The need for early diagnostic and prognostic markers of disease severity remains unmet; such markers could objectively guide decisions regarding the appropriateness of the level of care and the timing of initiation of life-saving supportive and therapeutic measures.

As for the immunopathogenesis of MIS-C, limited but key insights have emerged rapidly, most of which focus on the differences between MIS-C and KD. For example, Gruber et al. [12], and Consiglio et al. [13], showed that the inflammatory response in MIS-C differs from KD with respect to T cell subsets[13]. These conclusions were generally supported by two other studies, by Vella et al. [14], and Ramaswamy et al. [15] who also showed that severe MIS-C patients displayed skewed memory T cell TCR repertoires and autoimmunity characterized by endothelium-reactive IgG. Finally, Carter et al. [16], reported activation of CD4$^+$CCR7$^+$ T cells and γδ T cell subsets in MIS-C, which had not been reported in KD, which made them conclude that MIS-C may be a distinct immunopathogenic illness. While these studies further our understanding of MIS-C and the major conclusions of these studies are comprehensively reviewed elsewhere[17], it is noteworthy that each of these studies had some notable limitations— (i) in Gruber et al. [12], most of the MIS-C subjects were on immunomodulatory medications when samples were drawn; (ii) in Vella et al. [14], absence of contemporaneously analyzed healthy pediatric samples which were not available during the early phase of the pandemic; (iii) in Carter et al. [16], KD subjects were not concurrently studied and the authors themselves acknowledged that such side-by-side immunophenotyping of MIS-C and KD would be necessary to draw conclusions convincingly regarding similarities and differences between these two syndromes; (iv) absence of validation studies in independent cohorts in them all.

We recently showed that a 166-gene signature is conserved in all *vi*ral *p*andemics (ViP), including COVID-19, and a subset of 20-genes within that signature that classifies disease severity[18]. In the absence of a sufficiently large number of COVID-19 datasets at the onset of the COVID-19 pandemic, these ViP signatures were trained on only two datasets from the pandemics of the past (influenza and avian flu; GSE47963, $n = 438$; GSE113211, $n = 118$) and used without further training to prospectively analyze the samples from the current pandemic (i.e., COVID-19). The ViP signatures appeared to capture the 'invariant' host response, i.e., the shared fundamental nature of the host immune response induced by all viral pandemics, including COVID-19. Here we used the ViP signatures as a starting computational framework to navigate the syndrome of MIS-C that is still a relatively poorly understood entity, but none-the-less recognized as an immunologic response to viral exposure. More specifically, we sought to interrogate concurrently the quality and quantity of the shared and unique features in MIS-C and KD. Our results show that the nature of the host immune response in MIS-C is similar to that in the pre-pandemic syndrome of KD, i.e., both are characterized by a *IL15/IL15RA*-centric cytokine storm; however, MIS-C is farther along in the spectrum of disease severity.

## Results

**A gene signature seen in COVID-19 is also induced in KD, and tracks disease severity.** We sought to define the host immune response in KD and compare that to COVID-19 using an artificial intelligence (AI)-based approach. To this end, we took advantage of a recently identified analysis of the host immune response in COVID-19 in which over 45,000 transcriptomic datasets of viral pandemics were analyzed to extract a 166-gene signature[18] (summarized in Fig. 1a). Because publicly available transcriptomic datasets from SARS-CoV-2-infected patients were still relatively few at the onset of the pandemic, the rigor of analysis was increased through the use of an informatics approach, i.e., Boolean equivalent correlated clusters (BECC[19] Fig. 1a) that can identify fundamental invariant (i.e., universally conserved) gene expression relationships underlying any biological domain; in this case, the biological domain of '*respiratory viral pandemics*' was selected. Unlike some of the mainstream computational approaches (e.g., differential expression, Bayesian, and correlation network analyses, etc.) that are geared to identify the entire spectra of host immune response, BECC exclusively focuses on Boolean equivalent relationships to identify potentially functionally related gene sets that are part of the invariant spectrum of the host response. The resultant 166-gene ViP signature, developed from just two training datasets of pandemics of the past, was found to be conserved in all viral pandemics and outbreaks, including prospective studies on all COVID-19 datasets. The signature reflected the shared fundamental nature of the host immune response to multiple infectious triggers (Fig. 1b summarizes the types of pathogens that were found to induce the *ViP* signatures[18]). More specifically, the nature of the host immune response was found to be predominantly *IL15/IL15RA*-centric and enabled the formulation of precise therapeutic goals and measurement of therapeutic efficacy. At a molecular level, the ViP signatures were distinct from interferon-stimulated genes (ISGs[20,21]), in that, they revealed the broader and fundamental nature of the host immune response, shared between diverse pathogens and tissue/cell types. This included some tell-tale expected (Type I Interferon and cytokine signaling) and some unique (cellular senescence, exhaustion, chromatin silencing, regulation of apoptosis) pathway enrichments[18]. The latter, i.e., the unique pathways, were specifically enriched in a 20-gene subset of the ViP signature, which we called severe (s)ViP signature; this signature was trained on a large dataset of Influenza A/B-infected adult patients annotated with clinical severity[18]. The sViP signature predicted disease severity in COVID-19 (respiratory failure, need for mechanical ventilation, prolonged hospitalization and/or death)[18]. Consequently, the ViP signatures, but

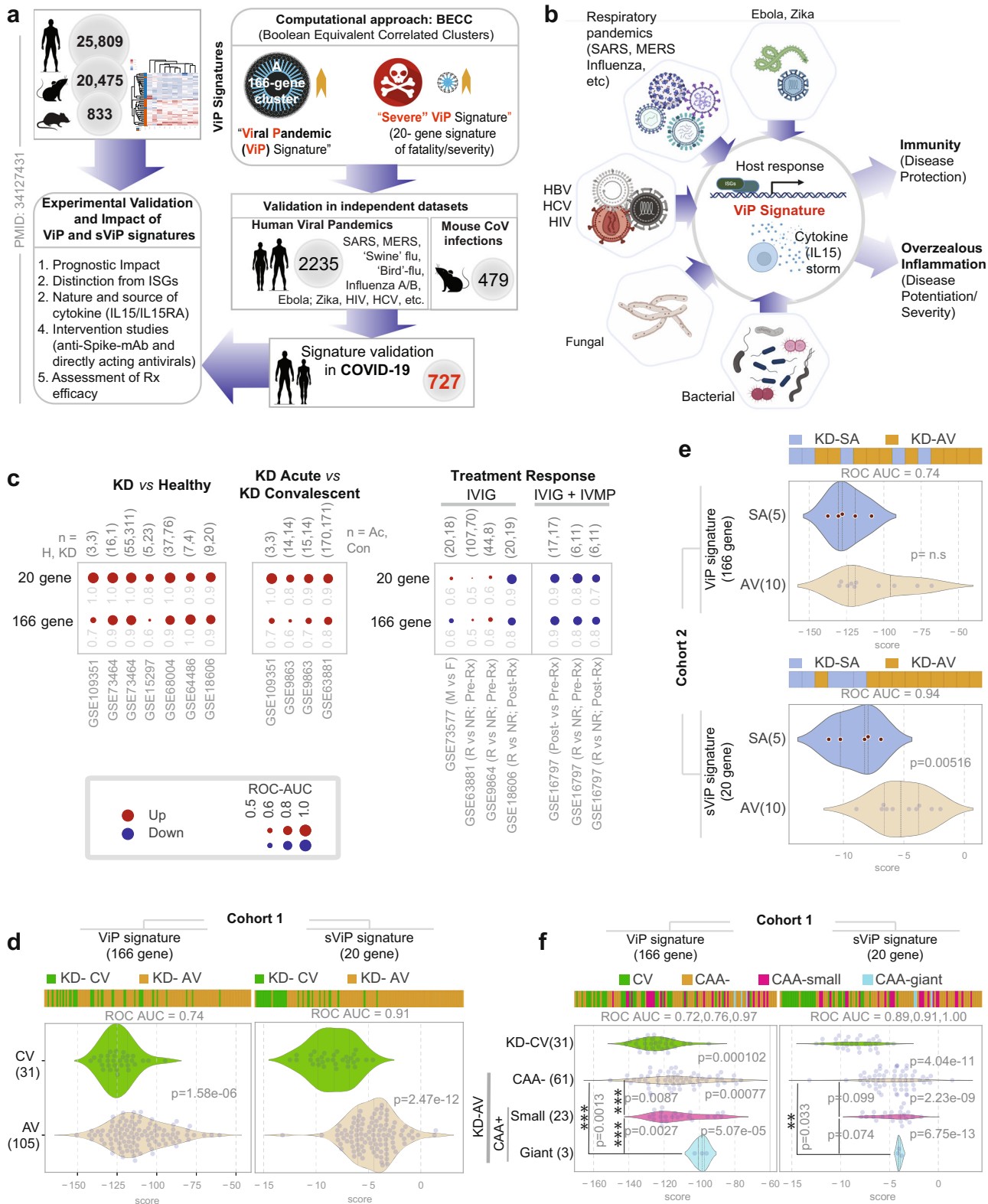

not ISGs, were found to be prognostic of disease severity in cohorts of COVID-19 datasets[18].

Because no KD and/or MIS-C samples were used previously in training the model that led to the discovery of the ViP signatures[18], we used these signatures as is, without further training, as quantitative and qualitative frameworks for measuring the immune response in publicly available historic cohorts of KD predating COVID-19. Both ViP and sViP signatures were

upregulated in blood and tissue samples derived from patients with KD compared to healthy controls (ROC AUC for classification of KD vs. healthy ranged from 0.8 to 1.00 across 7 independent cohorts; Fig. 1c, left), and that such induction was limited to the acute phase of KD and downregulated during convalescence (ROC AUC for classification of KD vs. healthy ranged from 0.6–0.8 for ViP and 0.8–1.00 for sViP across 4 independent cohorts; Fig. 1c, middle).

**Fig. 1 A Viral Pandemic (ViP) signature that is induced in COVID-19[18], is induced also in epidemic outbreaks of KD. a** Schematic displays the computational approach (BECC) and rigor (diversity and number of datasets) used to identify the 166-gene ViP and a subset of 20-gene severe (s)ViP signatures, and the subsequent experimentally validated inferences and impact of the same in a recent study[18]. The numbers in gray circles denote the total number of datasets analyzed in each category. **b** Schematic displays the various pathogenic triggers that induce ViP signatures (many of which are triggers also for KD) and the prominent induction of *IL15/IL15RA* as an invariant nature of the cytokine storm. **c** Bubble plots of ROC-AUC values (radii of circles are based on the ROC-AUC) demonstrating the strength of classification and the direction of gene regulation (Up, red; Down, blue) for the classification based on the 20-gene severe ViP signature (top) and 166-gene ViP signature (bottom) in numerous publicly available historic datasets. ViP signatures classified KD vs. healthy children (left), acute vs. convalescent KD (middle) and treatment response in the setting of combination therapy with IV steroids (MP methylprednisone) and IV IgG alone (IVIG), but not IVIG alone. Numbers on top of bubble plots indicate number of subjects in each comparison group. **d**, **e** Bar (top) and violin (bottom) plots display the classification of blood samples that were collected during acute (AV), sub-acute (SA; ~10-14 days post-discharge) and convalescent (CV; 1 year post-onset) visits from two independent KD cohorts (**d**; Historic Cohort 1; **e**; Prospective Cohort 2) using ViP (left) or sViP (right) signatures. **f** Bar (top) and violin (bottom) plots display the sub-classification of blood samples in Cohort 1 based on coronary artery aneurysm (CAA) status using ViP (left) or sViP (right) signatures. Welch's two sample unpaired two-sided *t*-test is performed on the composite gene signature score to compute the *p* values. In multi-group setting each group is compared to the first control group and only significant *p* values are displayed on the right. Additional pvalues are displayed on the left.

The strength of association between ViP/sViP signatures and acute KD was also preserved in datasets comprised of KD samples prospectively collected before and after IVIG treatment, and treatment response was annotated as responder (R) or non-responder (NR) (Fig. 1c, right). First, sex had no impact on the induction of signatures (ROC AUC 0.6 in Males vs. Females). Second, although the ViP/sViP signatures did not predict treatment response to IVIG (ROC AUC 0.5–0.6 in pre-treatment samples R vs. NR; GSE63881 and GSE9864), they were reduced in all responders compared to non-responders (ROC AUC 0.8–0.9 in post-treatment samples R vs. NR; GSE18606). Finally, in a study[22] in which the intervention was a combination of IVIG with the intravenous methylprednisolone (IVMP), both ViP signatures were reduced post-Rx (ROC AUC 0.9; GSE16797), and the signatures performed equally well in both pre-treatment and post-treatment samples in differentiating responders from non-responders (ROC AUC 0.7–0.8). These findings suggest that while the IVIG-IVMP combination regimen reduced the signatures effectively among all patients (*n* = 17), responders induced the ViP signatures to a lesser extent than non-responders. The 20-gene sViP signature consistently outperformed the 166-gene ViP signature in its ability to classify samples across all cohorts tested (Fig. 1c).

We next confirmed that both the ViP signatures are induced in acute KD (at presentation, ≤10 days of illness) compared to convalescent KD (day 289–3240 of Illness) in a large new cohort of consecutive patients (*n* = 105) who were diagnosed with the disease prior to the onset of the COVID-19 pandemic (Cohort 1; Supplementary Data 1) (Fig. 1d). Again, the sViP signature outperformed the ViP signature in sample classification (ROC AUC 0.91 vs. 0.74). In an independent cohort (Cohort 2, *n* = 20, Supplementary Data 1; Fig. 1e) prospectively enrolled in the current study after the onset of the COVID-19 pandemic, the ViP signatures could differentiate the acute from subacute (~10–14 d after discharge; ~day 17–25 of Illness) KD samples. As before, the 20-gene sViP signature outperformed the 166-gene ViP signature.

Finally, we tested the association between sViP signatures and markers of disease severity. Because CAA diameter is a predictor of coronary sequelae (thrombosis, stenosis, and obstruction)[23,24] and subsequent major adverse cardiac events (unstable angina, myocardial infarction, and death[25]), we used the development of coronary artery aneurysms (CAA) as a marker of disease severity. We found that both ViP signatures differentiated acute KD with giant CAAs (defined as a *z*-score of ≥10 or a diameter of ≥8 mm[26,27]) from convalescent KD samples (ROC AUC 0.95 and 0.97 for ViP/sViP signatures, respectively; Fig. 1f). The ViP signature effectively subclassified acute KD patients with giant aneurysms (CAA-giant) from to those with either no aneurysms

(CAA−; *p* value = 0.0027) or small aneurysms (CAA-small; *p* value = 0.0013). Similarly, the sViP signature effectively classified acute KD patients with giant aneurysms (CAA-giant) from those with no aneurysms (CAA−; *p* value 0.033). Such an analysis was not possible in Cohort 2 (Supplementary Data 1) because of the smaller cohort size and absence of subjects with giant CAAs.

We conclude that ViP signatures are induced in acute KD, and track disease severity, i.e., risk of developing giant CAAs, much like we observed previously in the setting of adult COVID-19[18]. Because ViP signatures represent the host immune response to diverse pathogens (Fig. 1b), upregulation of ViP signatures in KD is consistent with the hypothesis that KD is triggered by multiple infectious triggers[7,28,29], some of which may be viral in nature[30–32].

**Comparison of patients with MIS-C and Kawasaki disease.** Ten children were included who met the CDC definitions for MIS-C, with detectable anti-SARS-CoV-2 nucleocapsid IgG antibodies [Abbott Architect™] and undetectable virus by polymerase chain reaction (PCR; see Table 1). The MIS-C and KD cohorts had notable differences. Although sex and ethnicity were not different, the median age was higher (8.8 years) in the MIS-C cohort than in KD (Table 1), which is in keeping with our original report describing this syndrome in June 2020[2]. Left ventricular ejection fraction (LVEF) was reduced in the MIS-C cohort (*p* = 0.006), consistent with multiple prior reports[33–35]. While all patients had evidence of a marked inflammatory state, the MIS-C cohort had significant cytopenias, including low total WBC, absolute lymphocyte, absolute eosinophil, and platelet counts, with elevation of C-reactive protein level significantly above those observed in the KD cohort (Table 1). Most patients (90%) received intravenous immunoglobulin (IVIG) and 70% were treated with intravenous corticosteroids. One patient received anakinra, and three received infliximab. All patients made a full recovery. In all cases, blood was collected for serum before the initiation of any treatments.

**ViP/sViP signatures place MIS-C and KD on the same host immune continuum, but MIS-C as farther along the spectrum than KD.** We next analyzed whole blood-derived transcriptome and serum cytokine arrays in the current cohort of subjects with KD (Cohorts 2 and 4) and MIS-C (Fig. 2a). When MIS-C and acute KD groups were each compared to the control (subacute KD) samples, both ViP (Fig. 2b) and sViP (Fig. 2c) signatures were found to be induced at significantly higher levels in MIS-C samples compared to acute KD. However, when MIS-C and acute KD were compared to each other, we found that the ViP

| Table 1 Characteristics of patients with Kawasaki disease (KD) and MIS-C analyzed in this study. | | | | |
|---|---|---|---|---|
| Demographic and clinical parameters | KD | | MIS-C | p |
| | CAA− (n = 10) | CAA+ (n = 10) | MIS-C (n = 10) | |
| Age, yrs[a] | 2.2 (1.8-3.7) | 1.8 (1.2-3.5) | 8.8 (5.7-11.1) | 0.0002 |
| Illness day[b] | 6 (5-7) | 6 (5-7) | 4 (3-4) | NS |
| Male, n (%) | 6 (60) | 8 (80) | 6 (60) | NS |
| Ethnicity, n (%) | | | | |
| Asian | 1 (10) | 1 (10) | 0 | NS |
| African American | 1 (10) | 1 (10) | 2 (20) | |
| White | 3 (30) | 2 (20) | 1 (10) | |
| Hispanic | 4 (40) | 3 (30) | 6 (60) | |
| >2 races | 2 (20) | 5 (50) | 1 (10) | |
| Zmax | 1.0 (0.7-1.2) | 3.1 (2.7-3.3) | 1.9 (1.5-2.2) | 0.0002 |
| LVEF, median (IQR, range) | 67 (65-68, 56-76) | 70 (62-75, 56-79) | 58 (55-62, 31-65) | 0.006 |
| Lab data | | | | |
| WBC, 10³/μL | 12.6 (11.6-18.3) | 15.8 (13.0-18.5) | 11.1 (5.5-11.7) | 0.01 |
| ANC, /μL | 9159 (6904-106830 | 11319 (8896-12539) | 8172 (4437-10071) | NS |
| ALC, /μL | 2576 (1739-3658) | 2924 (2379-5075) | 939 (803-1019) | 0.0002 |
| AEC, /μL | 184 (104-529) | 378 (322-588) | 122 (0-231) | 0.04 |
| ZHgb | −1.4 (−2.3-0.1) | −1.5 (−1.8 to 0.6) | −2.2 (−2.7 to 0.7) | NS |
| PLT, 10³/mm³ | 330 (278-396) | 363 (338-396) | 177 (106-228) | 0.002 |
| ESR, mm/h | 55 (30-64) | 68 (58-76) | 44 (36-59) | NS |
| CRP, mg/dL | 6.1 (4.8-12.2) | 5.9 (4.0-8.0) | 21 (19.6-23.8) | 0.01 |
| BNP, pg/mL | ND | ND | 33 (19.7-23.6) | NA |
| Ferritin, ng/mL | ND | ND | 323 (223-960) | NA |
| Troponin, ng/mL | ND | ND | 0.02 (0.01-0.13) | NA |
| D-dimer, μg/mL | ND | ND | 1.94 (1.24-2.40) | NA |
| SARS-CoV-2 testing | | | | |
| PCR positive | ND | ND | 1 (10) | NA |
| IgG positive[c] | ND | ND | 10 (100) | NA |
| Treatment, n (%) | | | | |
| IVIG×1 | 10 (100) | 10 (100) | 8 (80) | NS |
| IVIG×2 | 0 | 1 (10) | 1 (10) | NS |
| Infliximab | 4 (40) | 10 (100) | 3 (30) | 0.003 |
| Anakinra | 0 | 1 (10) | 1 (10) | NS |
| Cyclosporine | 0 | 1 (10) | 0 | NS |
| Steroids | 0 | 0 | 7 (70) | 0.0001 |

Table displays the demographic, clinical and laboratory parameters collected on the KD and patients with MISC-C enrolled into the study. Zmax Maximum Z score (internal diameter normalized for body surface area) for the right and left anterior descending coronary arteries. Laboratory data are pre-treatment. Troponin was measured in nine patients with MIS-C. D-dimer was measured in eight patients with MIS-C. p-values were calculated by Kruskal–Wallis test for continuous variables among three groups and Chi-test for categorical variables.
LVEF left ventricular ejection fraction, WBC white blood cell count, PLT platelets, AEC absolute eosinophil count, ANC absolute neutrophil count, ALC absolute lymphocyte count, CRP C-reactive protein, ESR erythrocyte sedimentation rate, BNP brain natriuretic peptide, ZHgb hemoglobin concentration normalized for age, NS not significant, NA not applicable, ND not done.
[a]Median (Interquartile range (IQR) unless specified.
[b]Illness Day 1 = first day of fever.
[c]SARS-CoV-2 nucleocapsid IgG positive n = 9 and SARS-CoV-2 peptide array n = 1.

signatures could not distinguish between these samples, indicating that both conditions share a similar host immune response. Heatmaps of patterns of expression (Fig. 2d, e) demonstrate that most of the individual genes contributed to the elevated ViP and sViP signatures observed in MIS-C samples. These genes included *IL15* and *IL15RA* (highlighted in red; Fig. 2d), both components within the cytokine pathway that was previously demonstrated to be consistently elevated in the lungs of patients with fatal COVID-19 and in SARS-CoV-2 challenged hamsters[18].

Taken together, these analyses led to two key conclusions: (i) that the host immune response, as detected in a qualitative manner using the ViP signatures, is similar in KD and MIS-C and has a *IL15/IL15RA* shared component; (ii) that the degree of such host immune response, as measured quantitatively using the ViP signature scores, is more intense in MIS-C than KD. These findings are consistent with the fact that MIS-C is a host immune response to SARS-CoV-2 exposure, and we previously showed that the interaction of viral spike protein with the host entry receptor, ACE2 is critical for the induction of ViP signatures[18]. Findings are also in keeping with prior work[36] showing that serum levels of IL15 is significantly elevated in acute KD, ~10-fold

compared with subacute-KD and normal controls, and that such increase correlated with the concomitant increase in serum TNFα.

**A KD-specific signature independently confirms that KD and MIS-C are syndromes on the same host immune response continuum.** To circumvent an over-reliance on one set of signatures (i.e., ViP/sViP), we next analyzed a KD-specific 13 transcript diagnostic signature[37] that was previously shown to be effective in distinguishing children with KD from all other febrile conditions. During validation, the 13-transcript signature mirrored the certainty of clinical diagnosis, i.e., it differentiated definite, highly probable, and possible KD from non-KD with ROC AUCs of 98.1% (95% CI, 94.5–100%), 96.3% (95% CI, 93.3–99.4%), and 70.0% (95% CI, 53.4–86.6%), respectively (Fig. 2f). Unlike the ViP signatures, which has a typical enrichment of interferon and cytokine pathways with a prominent presence of *IL15/IL15RA*, the KD-signature is comprised of a set of non-overlapping genes, some of which relate to major central hubs within the tumor necrosis factor (TNFα) and interleukin 6

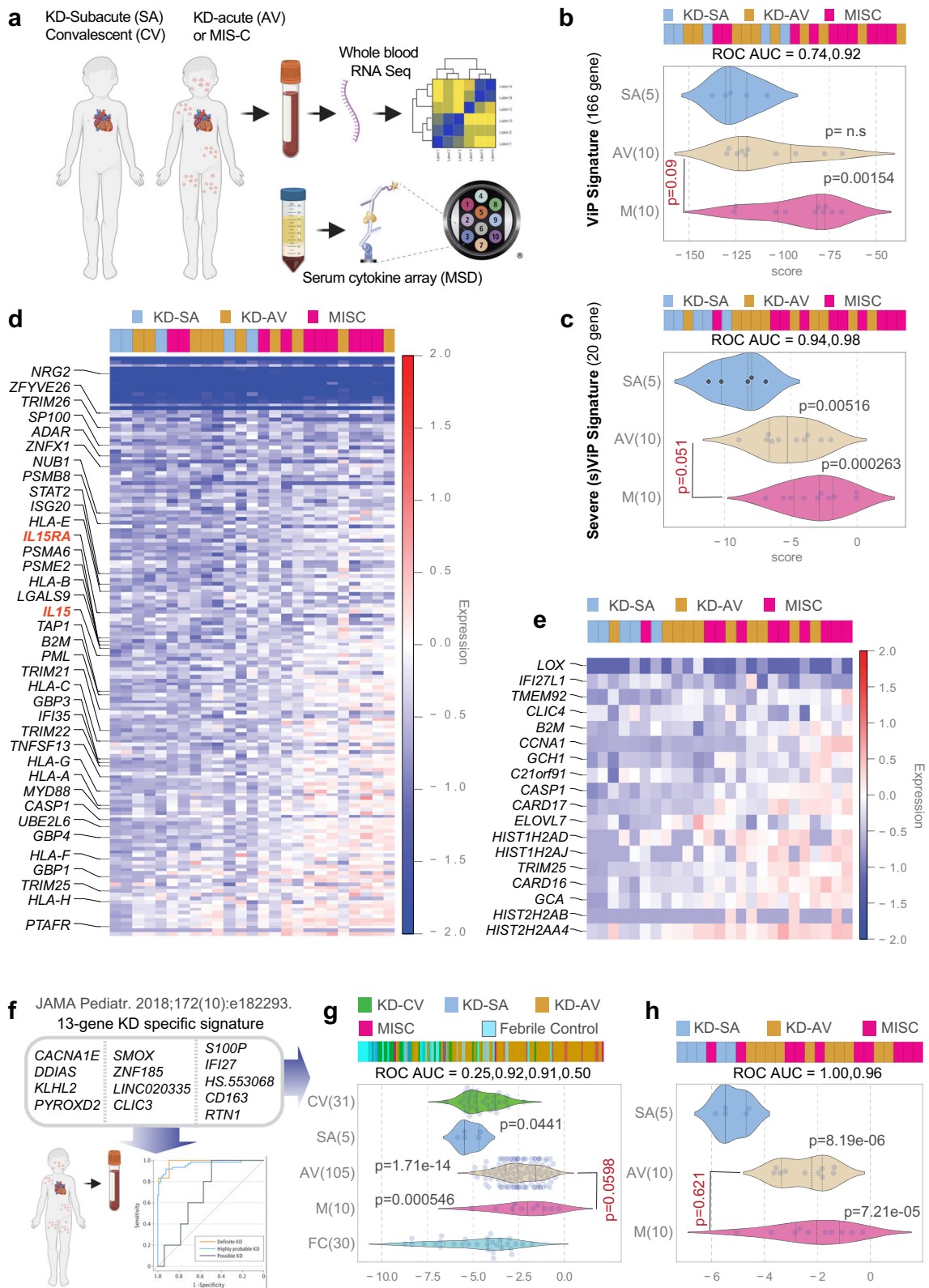

(IL6) pathways[37]. When we applied this signature to the historic cohort 1 (Fig. 2g) and to our current cohort (Cohort 2; Fig. 2h), we found that the KD-specific 13 transcript signature could not distinguish between MIS-C and KD in either cohort. Furthermore, a correlation test demonstrated that the two non-overlapping signatures, sViP and KD-13, both of which are significantly induced in KD and MIS-C (Fig. 2c) are independent of

each other (Supplementary fig. S1). This suggests that these two signatures reflect two fundamentally distinct and unrelated biological domains within the host immune response; whether their diagnostic/prognostic abilities may have an additive benefit remains to be explored.

The similar extent to which KD and MIS-C induced the KD-13 signature in two independent cohorts further supports our

**Fig. 2 A KD-specific 13 transcript signature[37] shows that KD and MIS-C are indistinguishable, but ViP/sViP signatures place MIS-C as farther along the spectrum than KD. a** Schematic displays the workflow for patient blood collection and analysis by RNA Seq (this figure) and cytokine array by mesoscale (Figs. 4 and 5). **b, c** Bar (top) and violin (bottom) plots display the classification of blood samples that were collected during collected during acute (AV) and sub-acute (SA; ~10–14 days post-discharge) visits of KD subjects and from patients diagnosed with MIS-C. The *p* value for comparison between acute KD (AV) and MIS-C (M) is displayed in red font. **d, e** Heatmaps display the patterns of expression of the 166 genes in ViP (**d**) and 20 gene sViP (**e**) signatures in the KD and MIS-C samples. The only cytokine–receptor pair within the signature, i.e., *IL15/IL15RA*, are highlighted on the left in red font in (**d**). **f** Schematic displays the 13-transcript whole blood signature (no overlaps with ViP signature genes) previously demonstrated to distinguish KD from other childhood febrile illnesses[37]. **g** and **h** Bar (top) and violin (bottom) plots display the classification of blood samples that were collected during acute (AV) and convalescent (CV) visits from two independent KD cohorts (**g**; Historic Cohort 1; **e**; Prospective Cohort 2) using 13-transcript KD signature. FC, febrile control. See also Supplementary Fig. S1 for co-dependence analysis of ViP and KD-13 signatures. Welch's two sample unpaired two-sided *t*-test is performed on the composite gene signature score to compute the *p* values. In multi-group setting each group is compared to the first control group and only significant *p* values are displayed. The *p* value for comparison between acute KD (AV) and MIS-C (M) is displayed in red font.

observation with ViP/sViP signatures that KD and MIS-C share fundamental aspects of host immune response with each other. That KD and MIS-C samples share ViP/sViP signatures with COVID-19 implies that the three diseases represent distinct clinical states on the same host immune response continuum.

**The sViP signature can recognize severe form of MIS-C that presents with myocardial dysfunction**. Next, we asked if the sViP signature can track disease severity in MIS-C. Because of the limited number of 'severe' cases in our MIS-C cohort, we prospectively analyzed two recently accessible MIS-C cohorts (GSE166489[15] and GSE167028[38]). While both datasets analyzed PBMCs from MIS-C subjects, and both studies used the presence of myocardial dysfunction as basis for severe disease, each study used a slightly different criterion for classification of disease severity (Fig. 3a). de Cevins et al.[38], classified MIS-C as severe when the patients presented with elevated cardiac troponin I and/or altered ventricular contractility by echocardiography, and clinical signs of heart failure requiring ICU support. Ramaswamy et al.[15], classified MIS-C as severe if they were critically ill, with cardiac and/or pulmonary failure. In both cohorts, sViP was able to classify severe MIS-C (with myocardial dysfunction; MYO+) from mild-moderate disease (who recovered or presented without myocardial dysfunction; MYO-) (Fig. 3b, c); while the *p* value was significant in GSE166489 (Fig. 3b), a similar trend was conserved in GSE167028[38] (Fig. 3c). These findings show that the sViP signature can identify severe MIS-C who are at risk to develop myocardial dysfunction, just as it did in the case of KD subjects who are at risk of developing giant CAAs (Fig. 1f) and similar to its prior performance in identifying adults with COVID-19 who are at risk of respiratory failure, mechanical ventilation, prolonged hospitalization and/or death[18].

Taken together with the prior findings, we conclude that the 20-gene sViP signature captures a core set of genes that are expressed in the setting of an overzealous (prolonged or intense, or both) host immune response in all three diseases—KD, MIS-C (this work) and COVID-19[18]—despite the fact that each present with distinct clinical features of severity.

Because all three conditions represent diseases of the immune system that share an 'infectious trigger', we asked if the ViP/sViP signatures are also induced in the setting of other diseases of the immune system. To this end, we analyzed numerous publicly available datasets, ranging from immunosuppressed states (as negative control), infectious diseases (both viral and bacterial; as positive control), and autoimmune diseases (Fig. 3d), and assessed the ability of ViP/sViP signatures to classify control and diseased samples in each dataset. Because the ViP/sViP signatures are able to detect the shared core fundamental host immune response in cell/tissue agnostic manner[18], we tested diverse samples ranging from whole blood to bronchoalveolar lavage fluid (Fig. 3d). The ViP/sViP signatures performed as

anticipated in the negative and positive control datasets, i.e., neither signature was induced in immunosuppressed conditions, e.g., malignancies, pregnancy, post-transplant immunosuppression, but both were induced in infectious diseases, e.g., sepsis, HIV, RSV, and tuberculosis (left; Fig. 3d). In the case of the autoimmune diseases, the ViP/sViP signatures were induced in some, but not others. The signatures were induced in those conditions that have multifactorial triggers, including potential contributions from infections; for example, mechanistic studies have identified viral link in many of them (EBV-linked autoimmune diseases[39] such as JIA, SLE, IBD). The signatures were not induced in other conditions where the disease triggers remain mysterious (e.g., sarcoidosis) or where the disease is driven by specific mutations, e.g., Neonatal onset multisystem inflammatory disease (NOMID) that is due to mutant NLRC3 and macrophage activation syndrome (MAS) that is due to mutant NLRC4). These findings lend further support to our finding that ViP/sViP signatures are induced and perform well to identify severe MIS-C, which shares infection as a trigger, much like KD and COVID-19. Intriguingly, numerous infectious and autoimmune diseases shared the *IL15/IL15RA*-centric cytokine response, which is in keeping with prior observations[40].

**Whole blood transcriptomes and cytokine panels reveal subtle differences between MIS-C and KD**. We next compared the whole blood transcriptomes from KD and MIS-C subjects (Fig. 3e) using a more conventional approach, which involved principal component analysis (PCA; Fig. 3f) in conjunction with hierarchical agglomerative clustering. The PCA analysis showed that 7 of 10 samples in both KD and MIS-C groups formed distinct clusters (circles in Fig. 3f), whereas 3 in each group were outside their respective clusters (yellow and cyan, Fig. 3f). Because agglomerative clustering is the most common type of hierarchical clustering used to group objects in clusters based on their similarity, we next sought this approach to assess grouping of KD and MIS-C samples (Fig. 3g) which also revealed two distinct clusters like PCA. Differential expression analysis was performed with the 7 MIS-C and 7 KD that formed distinct clusters in PCA and hierarchical clustering (Supplementary Data 3). Reactome pathway analyses of the genes upregulated in MIS-C revealed interferon and cytokine signaling (Fig. 3h, top), whereas the genes downregulated in MIS-C mostly enriched pathways concerning the complement cascade and phagocytosis, among others (Fig. 3h, bottom). When we analyzed the overlap between the 166-gene ViP signature and the up- or downregulated list of DEGs in MIS-C, we found that 11 genes overlapped between the ViP signature and upregulated genes in MIS-C (Fig. 3i). However, there was no overlap between the ViP signature and the downregulated genes. These analyses further emphasize the similarities between KD and MIS-C with more extreme features in MISC.

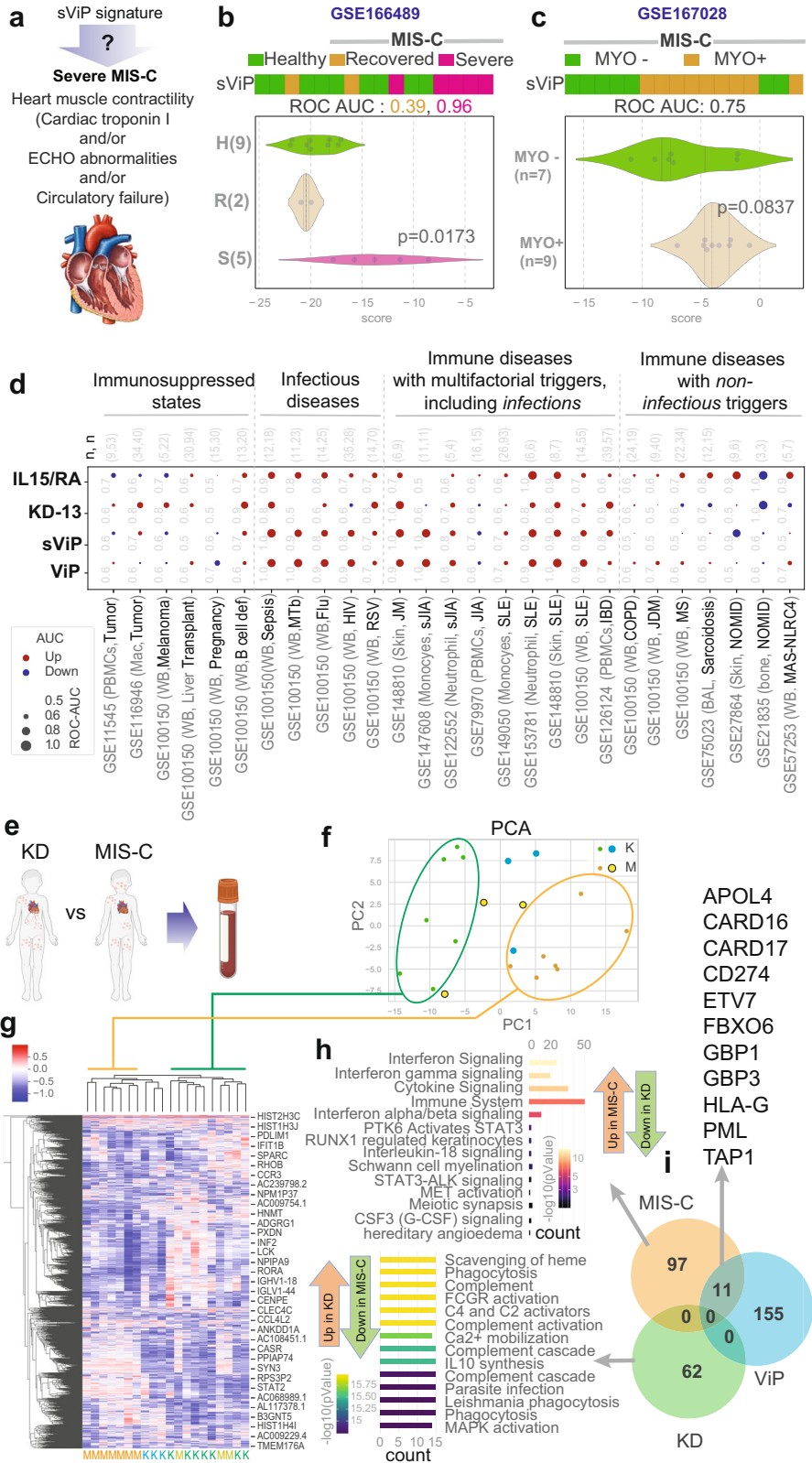

We next analyzed a set of 10 serum cytokines using meso scale discovery electrochemiluminescence (MSD-ECL) ultra-sensitive biomarker assay. A panel of 10 target cytokines was prioritized based on a review of the literature for the reported presence and/or relevance of each in either KD and/or MIS-C. An unsupervised clustering of just these 10 cytokines was sufficient to differentiate acute KD and MIS-C from one-year convalescent KD samples

(Fig. 4a; Cohorts #2 and #3, Supplementary Data 1, Supplementary Data 2); the convalescent samples served as baseline 'healthy' controls in this case. Regardless of their degree of elevation in the acute setting, all cytokines were virtually undetectable in convalescent samples (Supplementary Fig. S2, Supplementary Data 2). While most cytokines were induced indistinguishably in acute KD and MIS-C (Fig. 4a, b, top), notable exceptions were

**Fig. 3 Performance of ViP/sViP signatures on independent MIS-C datasets and on diverse tissues and in diverse diseases of the immune system.**
**a**–**c** Severe (s)ViP signature can classify severe MIS-C based on in two independent studies (GSE166489[15] and GSE167028[38]). Schematic in a summarizes the definition of severe MIS-C. **b**, **c** Classification of blood samples in two cohorts of MIS-C subjects, based on the need for ICU management due to the presence (MYO+) or recovery in the absence (R or MYO−) of myocardial dysfunction using sViP signature. Welch's two sample unpaired two-sided *t*-test is performed to compute the p values. **d** Bubble plots of ROC-AUC values (radii of circles are based on the ROC-AUC) and the direction of gene regulation (Up, red; Down, blue) in publicly available datasets using 4 gene signatures: the 166-gene ViP signature, the 20-gene sViP signature, the KD-13 signature, and finally the *IL15/IL15RA* composite score. Numbers on top of bubble plots indicate number (n) of control vs. disease samples in each dataset. Abbreviations: PBMCs peripheral blood mononuclear cells, Mac macrophages, WB whole blood, MTb *M. tubercutosis*, Flu Influenza, HIV human immunodeficiency virus, RSV respiratory syncytial virus, JM juvenile myositis, sjia systemic juvenile idiopathic arthritis, SLE systemic lupus erythematosus, IBD Inflammatory bowel disease, COPD chronic obstructive pulmonary disease, JDM juvenile dermatomyositis, MS multiple sclerosis, BAL bronchoalveolar lavage, NOMID neonatal onset multisystem inflammatory disease, MAS macrophage activation syndrome, NLRC4 NLR Family CARD Domain Containing 4. **e** Schematic showing the experimental design for studying differentially expressed genes (DEGs) in between KD and MIS-C subjects. **f**, **g** PCA (**f**) and a clustered heatmap analysis (**g**) of KD (green, **f**) and MIS-C (orange, **f**) samples are shown based on top 2242 genes according to mean absolute deviation identified using StepMiner algorithm[88]. Source data are provided. **h** Reactome pathway analysis of the DEGs between seven KD and seven MIS-C subjects in **f** (marked on the PCA). **i** Venn diagram between 166-gene ViP signature against the DEGs. Number of genes are indicated for each group in the Venn diagram. 11 overlapping genes between ViP signature and up-regulated in MIS-C are listed at the top.

TNFα, IFNγ, IL10, IL8 and IL1β, all of which were elevated to a greater extent in MIS-C compared to KD (Fig. 4b, bottom), either significantly (TNFα, IFNγ; Fig. 4b) or trended similarly, but failed to reach statistical significance (IL10, IL1β, IL8). Gene set enrichment analyses (GSEA) on the transcriptomic dataset for each of the differentially expressed cytokines (Fig. 4c) showed that the gene sets for those pathways were also induced in MIS-C at levels significantly higher than KD (Fig. 4d–h).

Taken together, these findings suggest that FDA-approved therapeutics targeting TNFα and IL1β pathways may be beneficial to treat MIS-C. The IL-1 receptor is expressed in nearly all tissues and its antagonism by anakinra, a recombinant form of IL-1Ra[41], prevents receptor binding of either IL-1α or IL-1β. Similarly, infliximab, a chimeric antibody to TNFα, has been repurposed for COVID-19[42–44], and our analyses suggest that this agent holds promise as a treatment for MIS-C.

**Integrated analyses of ViP/sViP signatures, cytokine profile, and clinical laboratory parameters reveal unique features of MIS-C and indicators of disease severity.** We next sought to understand how similar host cytokine responses can trigger two distinct clinical syndromes, and how such responses may drive features of clinical severity. To this end, we first carried out an agglomerative hierarchical clustering of the MIS-C and acute KD samples using both cytokine profiling (MSD) and clinical/laboratory parameters. This analysis, coupled with correlation tests (Supplementary Fig. S3) revealed several intriguing observations: (i) visualization by heatmap showed that compared to KD, MIS-C patients had higher cytokine levels and more severe pancytopenia (Fig. 5a); (ii) although platelet counts (PLT), but not absolute eosinophil counts (AEC) were significantly reduced in MIS-C compared to KD (Fig. 5b), there was a strong positive correlation between PLT and AEC in MIS-C, but not KD (Fig. 5c; left) and strong negative correlations of PLT with IL15 in both KD and MIS-C (Fig. 5c; right) and with MIP1α in MIS-C, but not KD (Fig. 5d); (iii) this is consistent with the fact that IL15 and MIP1α were found to have a strong positive correlation in MIS-C, but not KD (Fig. 5e). These findings suggest that MIS-C has key distinguishing features of thrombocytopenia and low eosinophil counts, and that both features are negatively correlated with the serum levels of IL15, a key feature of the ViP signature. These findings also held true when we analyzed the two clinical parameters, PLT and AEC, against ViP/sViP signatures, as well as a specific *IL15/IL15RA* composite transcript score from whole blood RNA Seq dataset. We found that PLT and AEC negatively correlated with ViP (Fig. 5f; top), sViP (Fig. 5f; middle) and a *IL15/IL15RA* composite score (Fig. 5f; bottom) in MIS-C, but

such correlations were restricted only to PLT in acute KD. These findings indicate that MIS-C, but not KD, has at least two distinct and interrelated clinical features, thrombocytopenia and eosinopenia, that appear to be related to the degree of induction of ViP signatures and a IL15-predominant cytokine induction. Findings also suggest that MIP1α is a key contributor to the immune response in MIS-C and that its levels are closely and positively related to the levels of IL15.

These findings reveal key similarities and differences among MIS-C, KD and COVID-19. Thrombocytopenia, which was more pronounced in MIS-C and correlated significantly with IL15 and *IL15/IL15RA* composite transcript score in both KD and MIS-C, has also been reported in COVID-19 and postulated because of various mechanisms[45–49]. In the case of KD, thrombocytopenia has been found to be associated with disease severity[50]. Similarly, in the case of COVID-19, a large meta-analysis confirmed that ~12% of hospitalized patients have thrombocytopenia, which represents a sign of disease severity and poor outcomes[45]. Thrombocytopenia carried a 3-fold enhanced risk of a composite outcome of intensive care unit admission, progression to acute respiratory distress syndrome, and mortality (odds ratio [OR], 3.49; 95% CI, 1.57–7.78), and a subgroup analysis confirmed a significant association with mortality (OR, 7.37; 95% CI, 2.08–26.14). Eosinopenia appears to be a notable shared feature between MIS-C and COVID-19[51], but not KD. These findings are consistent with the fact that KD is known to present with higher (not lower) eosinophil counts, Th2 cytokines IL-4, IL-5, and eosinophil cationic protein (ECP)[52–56]. As in the case of thrombocytopenia, persistent eosinopenia after admission correlated with COVID-19 severity and low rates of recovery[57].

**ViP/sViP signatures track the severity of two distinct cardiac phenotypes in MIS-C and KD.** We next analyzed the relationship between ViP signatures and the prominent and unique cardiac phenotype in MIS-C reported by others[33–35] and observed also in our cohort (Fig. 6a), i.e., a significantly reduced LVEF that can present with cardiogenic shock necessitating ionotropic support. We found that sViP signature scores, but not ViP or *IL15/IL15RA* composite scores correlate significantly with LVEF (Fig. 6b–d), indicating that LVEF may belong to the domain of clinical indicators of disease severity in MIS-C (alongside platelets and AEC), but it may not be directly related to the *IL15*-centric cytokine signaling. In KD, the ViP and sViP signatures were tested earlier (Fig. 1f) and found to distinguish patients with giant CAA from convalescent samples with ROC AUC > 0.95. A *IL15/IL15RA* composite score performed similarly in distinguishing those samples (Fig. 6e). We hypothesized that

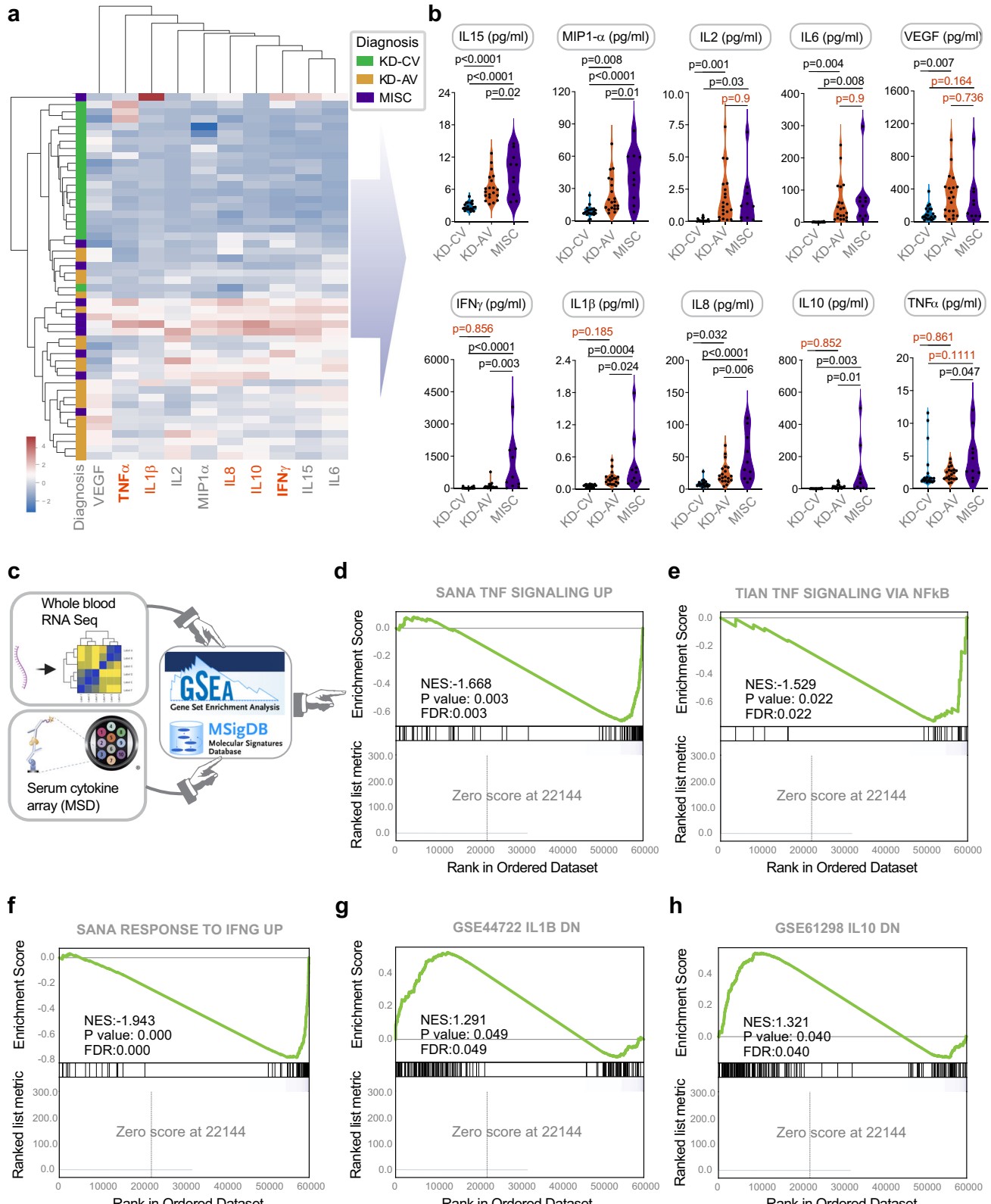

the ViP signatures may be related to two distinct cardiac phenotypes in severe disease: the signatures in KD may signify the nature of the vasculitis that drives the formation of CAAs, whereas the same signature in MIS-C may signify the degree of cardiomyopathy that impairs contractility (Supplementary Fig. S4a). Because we were unable to acquire cardiac tissues from MIS-C-related autopsies, we carried out immunohistochemical

analyses on cardiac tissues from a case of fatal KD. We found that both IL15 and IL15RA were expressed in the cardiomyocytes and coronary arterioles amidst extensive fibrosis, as detected using Masson's trichrome stain (Supplementary Fig. S4b).

Together, these findings suggest that the *IL15/IL15RA* induction we see in COVID-19, KD and MIS-C may have distinct sources and/or target end organs: We previously showed

**Fig. 4 Serum cytokine arrays and whole blood transcriptomes reveal the severity and nature of the cytokine storm in MIS-C that distinguishes it from KD. a** Heatmap displays the results of unsupervised clustering of sub-acute and acute KD (KD-SA, KD-AV; $n = 10$ each) and MIS-C ($n = 10$) subjects using the cytokine profiles determined by mesoscale (MSD). Red = cytokines differentially expressed between MIS-C and KD. See also Supplementary Fig. S2 for violin plots for individual cytokines. **b** Source data are provided as a Supplementary Data 2. Violin plots display the shared (top panels; IL15, MIP1a, IL2, IL6 and VEGF) and distinct (bottom panels; IFNγ, IL1β, IL8, IL10, and TNFα) features of the cytokine storm in MIS-C vs. KD subjects. Statistical significance was determined by one-way ANOVA followed by Tukey's test for multiple comparisons. **c** Schematic shows the process used to integrate serum cytokine array results with whole blood RNA Seq data; cytokines that were differentially expressed in MIS-C were used to inform GSEA of the corresponding pathways. **d–f** Gene set enrichment analysis (GSEA pre-ranked analysis) of three pathways derived from MSigDB: SANA TNF SIGNALING UP (**d**), TIAN TNF SIGNALING VIA NFkB (**e**), and SANA RESPONSE TO IFNG UP (**f**) demonstrate the significance of TNF (**d**, **e**) and IFNG (**f**) pathway activation in MIS-C. **g**, **h** Down-regulated genes after IL1B (**g**) and IL10 (**h**) stimulation were derived from differential expression analysis of GSE44722 ($n = 269$ genes), and GSE61298 ($n = 208$ genes) respectively. GSEA pre-ranked analysis to test the significance of IL1B and IL10 pathway is performed like panels **d–f** using the down-regulated genes. GSEA pre-ranked analysis computes nominal pvalue and FDR using an empirical phenotype-based permutation test procedure. No adjustments were made for multiple comparisons because of single hypothesis testing. Source data are provided as a Source Data file.

prominent induction of *IL15/IL15RA* in the lung alveoli of fatal COVID-19 patients[18], and here we show it in the coronary arteries and cardiomyocytes in KD. However, the latter claim has some notable caveats: (i) it is based on one fatal KD and no MIS-C tissue, hence, may not be representative of what happens in every KD heart; (ii) the induction of *IL-15/IL-15R* may be consistent with systemic hyperinflammation, but immune cell infiltration in the heart was not evident; (iii) because IL-15 has been implicated in a wide range of cardiovascular diseases[58] and was previously shown to be implicated in cardiomyocyte survival[59] during oxidative stress, alternative explanations other than hyperinflammation could be at play. Thus, further studies are required to determine if the findings in KD and MIS-C can be generalized and to determine what may be the impact of *IL-15/IL-15R* expression in the heart.

## Discussion

Using a combination of publicly available KD datasets and newly recruited cohorts of KD and MIS-C subjects (summarized in Fig. 7a) and a set of gene signatures we report an unexpected discovery regarding the host immune response in these diagnoses. Our findings show that two distinct clinical syndromes, KD, which predates the current pandemic by 6 decades, and the novel COVID-19, share a similar profile of host immune response. The same host immune response is seen also in MIS-C, a new disease that co-emerged with COVID-19, which has some overlapping features with KD (i.e., clinical presentation, pediatric, etc.), and yet is an immune response to the virus that causes COVID-19 (Fig. 7b). We assessed the quality and intensity of the host immune response in these syndromes with a powerful and unbiased computational tool, the ViP signatures[18]. Challenging to our previous understanding of MIS-C as post-infectious syndrome, recent studies have revealed a prolonged presence of viral replication[60] and dendritic cell exhaustion caused by the persistence of the antigen[61]. The use of this computational tool was rationalized because MIS-C is triggered by exposure to a virus and therefore, the induction of the ViP signatures in the acute phase, followed by their reduction during convalescence was anticipated. Previously we had demonstrated the usefulness of the ViP signatures to define and measure the host immune response in COVID-19, identify the site/source of the cytokine storm, track disease severity, objectively formulate therapeutic goals and track the effectiveness of emerging drugs/biologics[18]. We now show that the same ViP signatures can objectively demonstrate the shared immunophenotypes between all three syndromes (COVID-19, KD and MIS-C), which features an upregulation of the *IL15/IL15RA* pathway. That a 13 transcript KD-specific signature that was previously shown to distinguish KD from other non-KD febrile illnesses[37] failed to distinguish KD from MIS-C, further confirmed that KD and MIS-C share similar molecular

markers of disease and hence, are fundamentally similar at the molecular level (summarized in Fig. 7b). Findings were confirmed also using conventional approaches (PCA and hierarchical agglomerative clustering followed by DEG analyses). Taken together, these results are in keeping with what has been observed by Consiglio et al.[13], who found KD and MIS-C to be clustered together in a PCA analysis of plasma proteins. These findings suggest that the two clinical syndromes not just share common clinical features (e.g., rash, fever, etc.), but may also share proximal pathways of immunopathogenesis.

Despite the high-level broad similarities, the ViP signatures also helped identify key differences in clinical/laboratory parameters that may help distinguish MIS-C from KD. First, although the ViP signatures placed KD and MIS-C on the same host immune continuum, the degree of host immune response in MIS-C is significantly higher than KD by all measures tested (i.e., ViP, sViP, *IL15/IL15RA* and KD-13 signatures and direct measurement of serum cytokines). Higher ViP signatures in MIS-C tracked three major clinical and/or laboratory parameters (see Fig. 7b): (i) degree of thrombocytopenia in severe cases (all three diseases); (ii) eosinopenia (in COVID-19 and MIS-C, but not KD) and (iii) impaired cardiac contractility (unique to MIS-C; but not KD); (iii) an integrated analysis of serum cytokines and transcriptomics revealed that the proinflammatory MIP1α, TNFα, and IL1 pathways are significantly induced in MIS-C compared to KD. In light of these findings, a rational approach to MIS-C treatment would be to combine the FDA-approved drugs anakinra[41] and infliximab[42–44]. In fact, during the preparation of this manuscript a new study has already shown favorable outcome in MIS-C with the use of Infliximab[62]. Furthermore, our findings are consistent with the recently released guidelines by the American College of Rheumatology for initial immunomodulatory treatment of MIS-C[63]; it is noteworthy that these guidelines were released while this work was under review.

Finally, our findings reveal a pattern of MIS-C-defining molecular features (*IL15/IL15RA*, MIP1α, *TNFα*, and *IL1* pathways) and clinical and laboratory parameters (thrombocytopenia, eosinopenia, and reduced myocardial function). For example, MIP1α elevation shows strong correlations with clinical features of disease (low PLT, high IL15 and low AEC) in MIS-C, but not KD. This suggests two things—(i) that despite shared proximal proximal pathways of immunopathogenesis (i.e., *IL15/IL15RA*-centric cytokine storm), the immunopathogenesis of KD and MIS-C may diverge distally; and (ii) that *IL15/IL15RA*, eosinopenia and thrombocytopenia may be inter-related phenomena in the setting of infection and inflammation. Platelets, besides their role in hemostasis, they are known to participate in the interaction between pathogens and host defense[64–66]. Persistent thrombocytopenia carried higher mortality in sepsis[67,68], and in COVID-19[69,70]. Our analysis revealed a direct and unusually

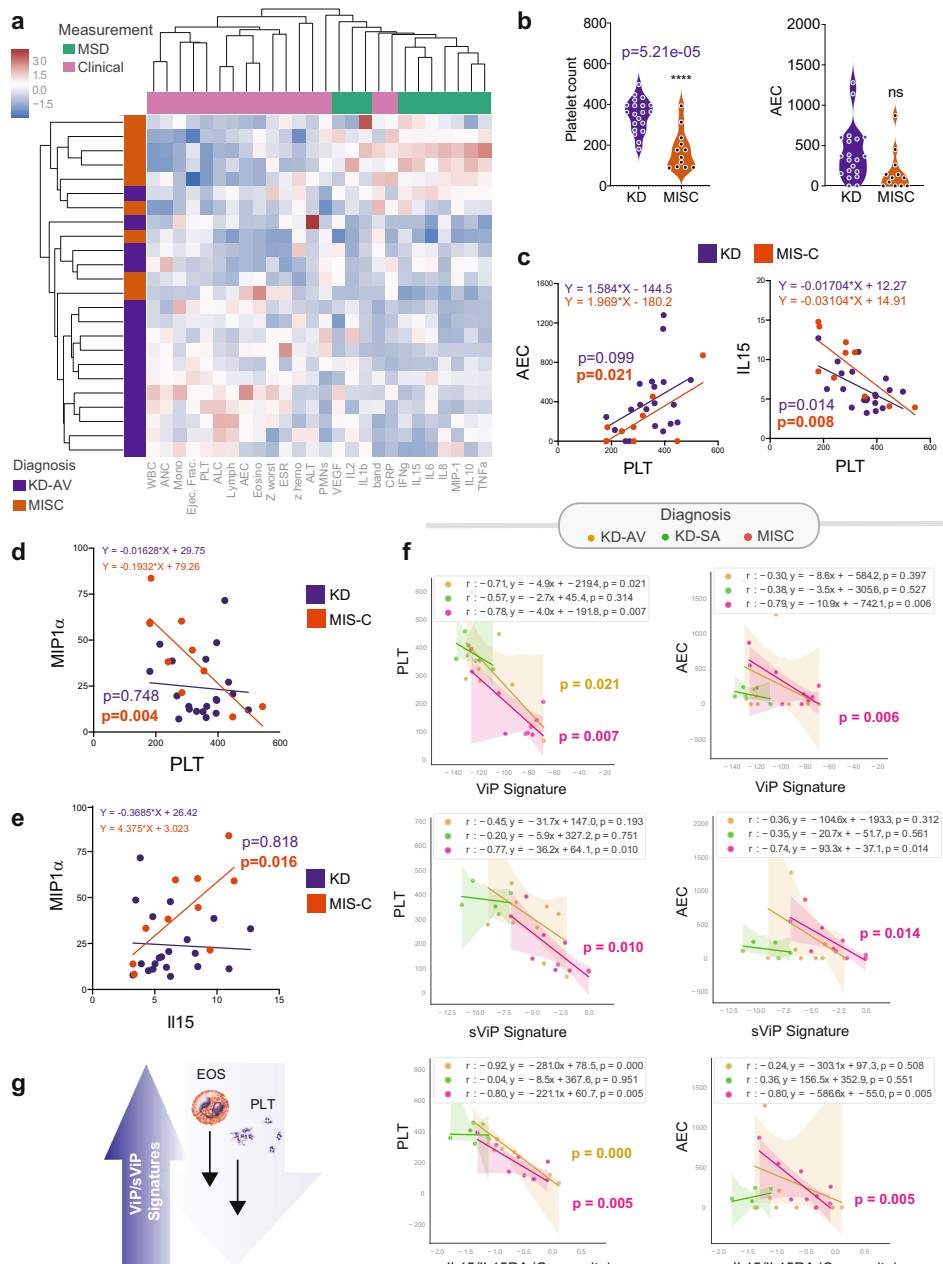

**Fig. 5 An integrated analysis of mesoscale (cytokine) data, ViP/sViP transcriptomic signatures and laboratory and clinical parameters reveals features that are unique to MIS-C. a** Heatmap displays the results of hierarchical agglomerative clustering of acute KD (KD-AV; $n = 10$) and MIS-C ($n = 10$) subjects using the cytokine profiles determined by mesoscale (MSD) and the laboratory features. Source data are provided as a Source Data file. **b** Violin plots display PLT (platelet) and AEC (absolute eosinophil counts) in KD and MIS-C (unpaired two-sided Student's *t*-test used to test significance). **c–e** Correlation test (two-sided test of the slope of the regression line compared to zero) between AEC and PLT (**c**; left) and IL15 and PLT (**c**; right), and MIP1α and PLT (**d**) and MIP1α and IL15 (**e**) are shown, and significance was calculated and displayed using GraphPad Prism 9. Significance: ns: non-significant, ****$p < 0.0001$. See Supplementary Fig. S3 for all possible correlation tests between clinical and cytokine data in KD, MIS-C and COVID-19. **f** Correlation tests between PLT (left) or AEC (right) on the *Y*-axis and gene signature scores on the *X*-axis [either ViP (top), sViP (middle) or a *IL15/IL15RA* composite (bottom)] were calculated and displayed as scatter plots using python seaborn lmplots with the *p*-values. The confidence interval around the regression line is indicated with shades. **g** Schematic summarizing the findings in MIS-C based on laboratory and RNA seq analysis.

strong correlation between thrombocytopenia and eosinopenia in MIS-C. Eosinophils, on the other hand, as reviewed elsewhere[71], have important antiviral properties, attributed to their granular proteins (e.g., eosinophil-derived neurotoxin, cationic protein) that display antiviral activities against single-stranded RNA viruses. Eosinophils can also support viral clearance[72]. Eosinopenia, in the setting of acute infection, has been found to be a

direct response to infectious stimuli[73], TLR4 ligands and chemotactic factors[74], and has been considered a reliable diagnostic marker of infection[75] in critically ill patients and a predictor of mortality[75–77]. Of relevance to the pediatric syndrome MIS-C, eosinopenia is encountered in about a 1/3rd of the pediatric COVID-19 subjects[78]. It is noteworthy that eosinopenia (defined as an eosinophil count <15 cells/μL and an eosinophil

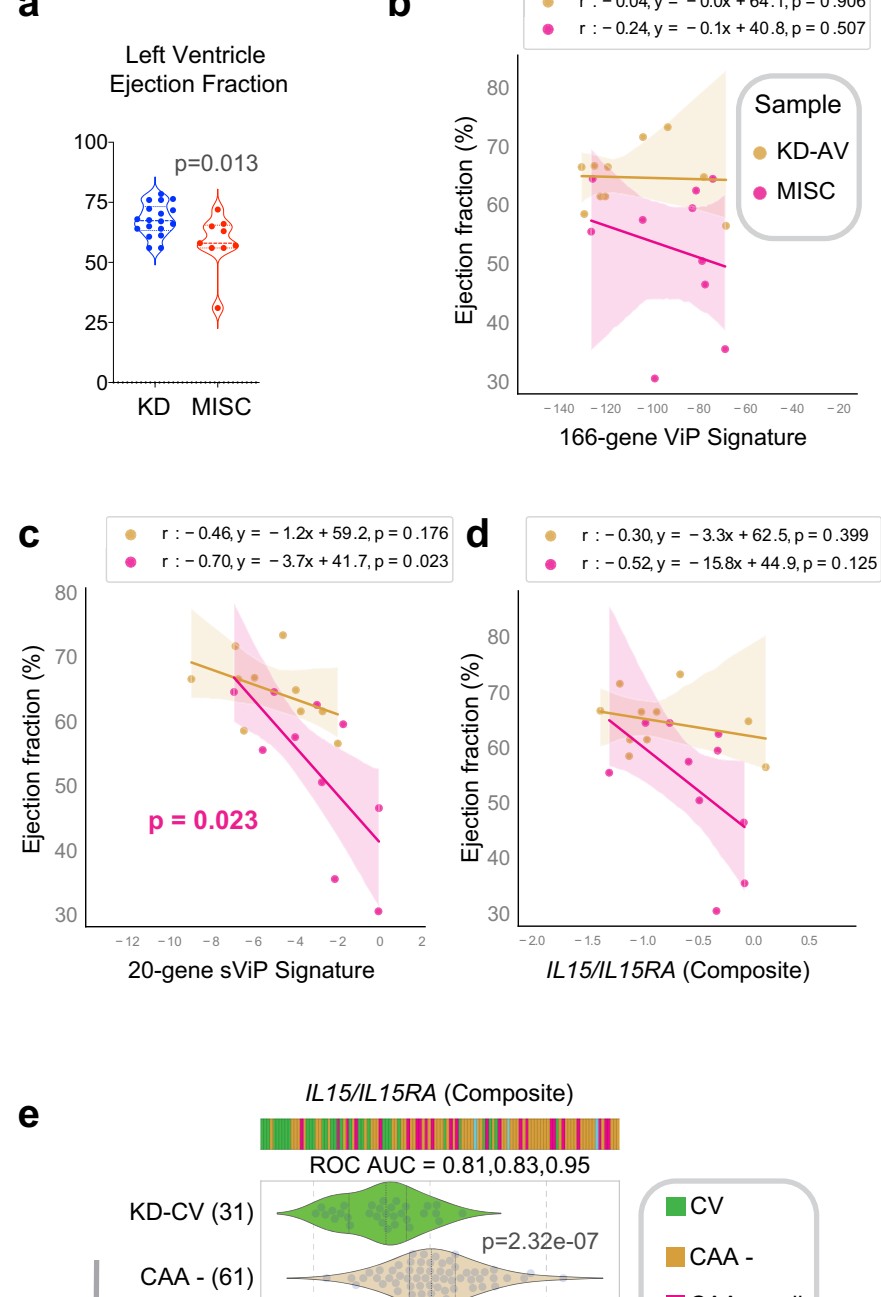

**Fig. 6 ViP/sViP signatures correlate with two distinct cardiac phenotypes in MIS-C and KD. a** Violin plots display the left ventricular ejection functions (LVEF) in KD and MIS-C patients. Statistical significance was determined by unpaired two-sided Student's *t*-test. **b**–**d** Correlation tests (two-sided test of the slope of the regression line compared to zero) between LVEF (*Y*-axis) and gene signature scores on the *X*-axis [either ViP (**b**), sViP (**c**), or a *IL15/IL15RA* composite (**d**)] are displayed as a scatter plot and significance was calculated and displayed as in Fig. 5f. The confidence interval around the regression line is indicated with shades. **e** Bar and violin plots show how a *IL15/IL15RA* compositive score varies between KD samples. The score classifies KD-AV with giant CAAs from control (KD-CV) samples with a ROC AUC 0.95. Welch's two sample unpaired two-sided *t*-test is performed on the composite gene signature score to compute the *p* values. In multi-group setting each group is compared to the first control group and only significant *p* values are displayed.

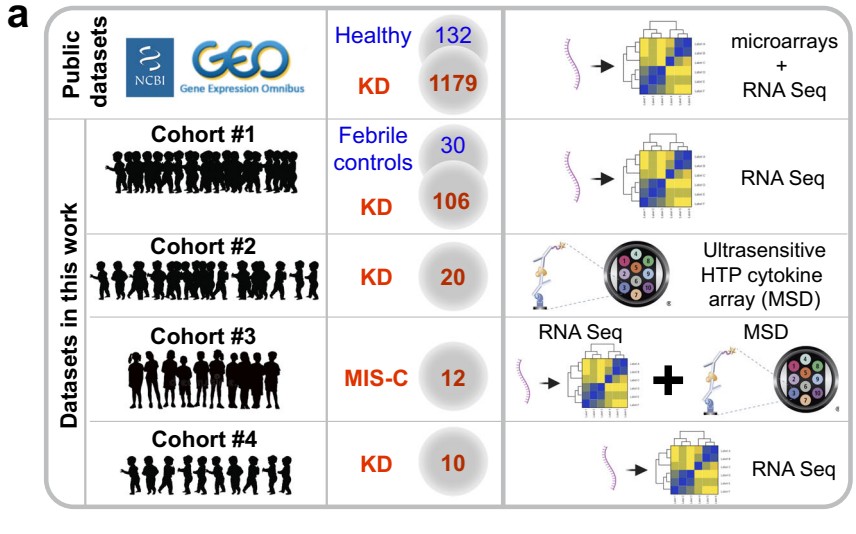

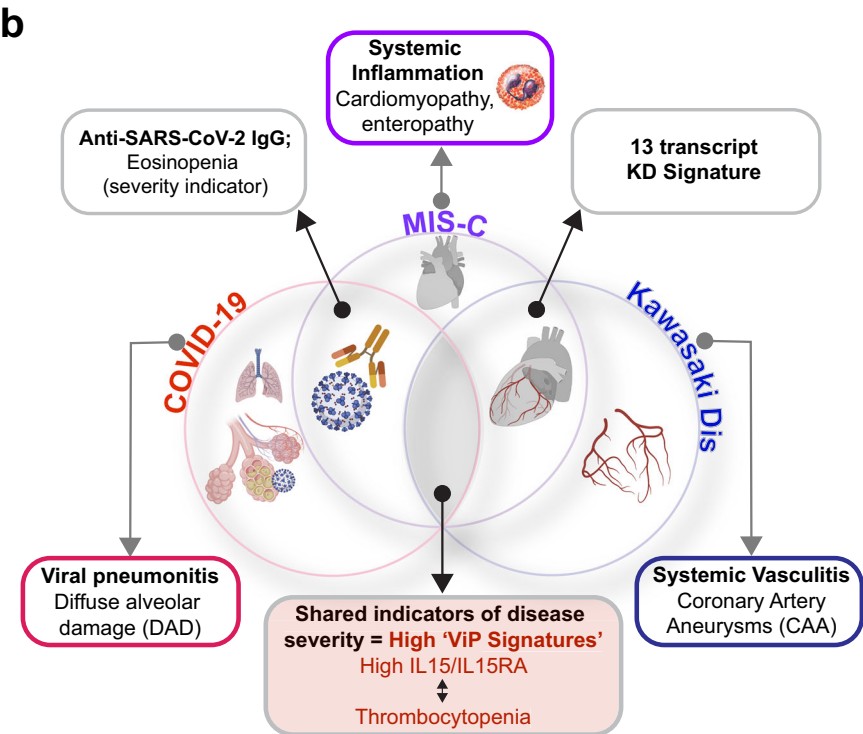

**Fig. 7 Summary of findings and conclusions. a** Summary of datasets used (publicly available prior ones and new original cohorts) to support the conclusions in this work. Numbers in circles denote the number of subjects in each cohort. **b** Venn diagram displays the major findings from the current work. ViP/sViP signatures, and more specifically, the *IL15/IL15RA* specific gene induction are shared between patients in all three diagnostic groups. While these signatures are known to be associated with diffuse alveolar damage in the lungs of patients with COVID-19[18], it is associated with CAA in KD and with reduction in cardiac muscle contractility in MIS-C. Overlapping features between each entity are displayed.

percentage < 0.25%) is a known poor prognostic factor for admissions into the pediatric ICU (hazard ratio [HR]: 2.96; $P = 0.008$[79]). It is possible that the two related clinical/laboratory parameters (low PLT and AEC) may be useful indicators of disease severity and prognosis in MIS-C and may guide decision-making in therapy and level of care in the hospital setting.

The strength of our study lies in the concurrent analysis of KD and MIS-C samples, our access to relatively large and independent cohorts of patients (in the case of KD), our ability to include age-matched pediatric healthy controls and febrile controls (non-KD and non-MIS-C, both pre-pandemic), and that all samples were drawn prior to the initiation of treatment. In doing so, this study overcomes some of the limitations of prior studies[12,14,16]. Another strength is the use of a set of *ViP* signatures (that were validated in COVID-19)[18] and a KD-diagnostic signature[37] as the computational framework to compare the two syndromes. Last, but not the least, the multi-omics approach used here on samples obtained from the same patients allowed us to predict and validate the prominent upregulation of one shared cytokine pathway (i.e., *IL15*) at both transcript and protein level. Notable limitations of our study include a relatively small sample size of MIS-C subjects ($n = 12$), limited number of publicly available MIS-C datasets for independent validation, and our inability to access cardiac tissue from KD and MIS-C subjects. Future studies on

emerging datasets will enable rigorous validation of the analysis presented here.

## Methods

**Kawasaki disease (KD), multisystem inflammatory syndrome in children (MIS-C), febrile control (FC) subjects.** All KD subjects met the American Heart Association (AHA) criteria[80] for complete or incomplete KD and subjects in this study were enrolled before the SARS-CoV-2 pandemic. Demographic and clinical data including echocardiography data and laboratory values were prospectively collected and entered into an electronic database. Coronary artery Z-scores were classified according to the AHA 2017 guidelines as follows: normal <2.0; dilated, $2 \leq Z < 2.5$; aneurysm: $2.5 < Z < 10.0$; and giant aneurysm $\geq 10.0$.

All MIS-C patients met the case definition from the Centers for Disease Control and Prevention. Subjects were enrolled prospectively with collection of acute, pre-treatment samples. Demographic and clinical data including echocardiography data and laboratory values were prospectively collected and entered into an electronic database.

Febrile control patients had fever of at least three days duration and at least one mucocutaneous feature of KD including rash, conjunctival injection, or mucosal erythema. All were enrolled prior to the onset of the pandemic.

The final diagnosis for the control patients was adjudicated by a pediatric infectious disease specialist (J.C.B.) and by a pediatric emergency room physician (J.K.) at least 2 months after initial presentation when testing results and clinical outcome were known. The final diagnoses of the 30 FC were defined by PCR or viral culture and included the following infections: 12 adenovirus, 5 EBV, 2 metapneumovirus, 3 rhinovirus, 3 influenza, 2 parainfluenza, 2 RSV, and 1 measles.

The characteristics of patient cohorts that were part of this study are included in Supplementary Data 1. The study protocol was reviewed and approved by the institutional review board at UCSD (UCSD # 14020). Written informed consent from the parents or legal guardians and assent from patients were obtained as appropriate. For all the deidentified human subjects, information including age, gender, and previous history of the disease, was collected from the chart following HIPAA guidelines. The study design and the use of human study participants was conducted in accordance with the criteria set by the Declaration of Helsinki. Patients were not compensated for their participation in this study.

**Collection of blood samples and RNA isolation.** Whole blood was collected into PAXgene® tubes (PreAnalytiX) for RNA and into red top tubes for serum before the initiation of any treatment (illness day ≤10) for the KD, MIS-C and FC subjects (illness day <15 for some FC subjects) and at the clinic visit (day 17–25 of Illness for subacute and day 289–3240 of Illness for late convalescent) for the KD subjects. RNA was extracted following manufacturer's instruction (PAXgene Blood miRNA Kit). Serum was separated immediately by centrifugation and stored at −80 °C until use.

**Tissue samples.** We obtained formalin-fixed, paraffin-embedded tissues from a 4-year-old female who died 9 months after KD onset due to thrombosis of giant aneurysms. Written consent was obtained from the parents. The tissue sampling protocol was reviewed and approved by the Institutional Review Board at UCSD (UCSD# 180587).

**ViP and severe (s)ViP signatures.** ViP (Viral Pandemic) signature is derived from a list of 166 genes using Boolean Analysis of large viral infection datasets (training datasets: GSE47963, $n = 438$; GSE113211, $n = 118$). This 166-gene signature was conserved in all viral pandemics, including COVID-19, inspiring the nomenclatures ViP signature[18]. A subset of 20-genes classified disease severity called severe-ViP signature using an additional cohort (GSE101702, $n = 159$)[18]. To compute the ViP signature, first the genes present in this list were normalized according to a modified Z-score approach centered around StepMiner threshold (formula = (expr-SThr)/3*stddev). The normalized expression values for every probeset for 166 genes were added together to create the final ViP signature. The severe ViP signature is computed similarly using 20 genes. The samples were ordered finally based on both the ViP and severe-ViP signature. A color-coded bar plot is combined with a violin plot to visualize the gene signature-based classification.

**Data analysis.** Several publicly available microarrays and RNASeq databases were downloaded from the National Center for Biotechnology Information (NCBI) Gene Expression Omnibus (GEO) website[81–83]. Gene expression summarization was performed by normalizing Affymetrix platforms by RMA (Robust Multichip Average)[84,85] and RNASeq platforms by computing transcripts per millions (TPM)[86,87] values whenever normalized data were not available in GEO. We used log2(TPM + 1) as the final gene expression value for analyses. GEO accession numbers are reported in figures, and text. KD/MIS-C RNASeq datasets were processed using salmon. Batch correction was performed using ComBat_seq R package.

**StepMiner analysis.** StepMiner is a computational tool that identifies step-wise transitions in a time-series data[88]. StepMiner analysis is used to identify the threshold to convert continuous gene expression values into Boolean values (High/Low). StepMiner performs an adaptive regression scheme to identify the best possible step up or down based on sum-of-square errors. The steps are placed between time points at the sharpest change between low expression and high expression levels, which gives insight into the timing of the gene expression-switching event. To fit a step function, the algorithm evaluates all possible step positions, and for each position, it computes the average of the values on both sides of the step for the constant segments. An adaptive regression scheme is used that chooses the step positions that minimize the square error with the fitted data. Finally, a regression test statistic is computed using Eq. (1)

$$F\,stat = \frac{\sum\limits_{i=1}^{n}(\hat{X}_i - \bar{X})^2/(m-1)}{\sum\limits_{i=1}^{n}(X_i - \hat{X}_i)^2/(n-m)} \qquad (1)$$

where $X_i$ for $i = 1$ to $n$ are the values, $\hat{X}_i$ for $i = 1$ to $n$ are fitted values. m is the degrees of freedom used for the adaptive regression analysis. $\bar{X}$ is the average of all the values: $\bar{X} = \frac{1}{n} * \sum_{j=1}^{n} X_j$. For a step position at $k$, the fitted values $\hat{X}_l$ are computed by using Eq. (2)

$$\frac{1}{k} * \sum_{j=1}^{k} X_j \text{ for } i = 1 \text{ to } k \text{ and } \frac{1}{(n-k)} * \sum_{j=k+1}^{n} X_j \text{ for } i = k+1 \text{ to } n \qquad (2)$$

**Boolean analysis.** Boolean logic is a simple mathematic relationship of two values, i.e., high/low, 1/0, or positive/negative. The Boolean analysis of gene expression data requires the conversion of expression levels into two possible values. The *StepMiner* algorithm is reused to perform Boolean analysis of gene expression data[89]. The Boolean analysis is a statistical approach which creates binary logical inferences that explain the relationships between phenomena. Boolean analysis is performed to determine the relationship between the expression levels of pairs of genes. The *StepMiner* algorithm is applied to gene expression levels to convert them into Boolean values (high and low). In this algorithm, first the expression values are sorted from low to high and a rising step function is fitted to the series to identify the threshold. Middle of the step is used as the StepMiner threshold. This threshold is used to convert gene expression values into Boolean values. A noise margin of 2-fold change is applied around the threshold to determine intermediate values, and these values are ignored during Boolean analysis.

**Boolean equivalent correlated clusters (BECC) analysis.** BECC analysis[19] is based on Boolean Equivalent[89] relationships, pair-wise correlation and linear regression analysis. BECC analysis identified ViP and severe-ViP signature using the BooleanNet statistic[18].

**Heatmaps, hierarchical agglomerative clustering, PCA, differential expression analysis.** Gene expression values were normalized according to a modified Z-score approach centered around *StepMiner* threshold (formula=(expr- SThr)/3*stddev). The samples were ordered according to average of the normalized gene expression values in the largest cluster along the Boolean path. The heatmap use red colors for the high values, white colors for the intermediate values and blue colors for low values. Gene names for few selected genes are highlighted on the left to show their expression patterns. Rows and columns are ordered based on hierarchical agglomerative clustering using python seaborn (version 0.10.1) clustermap function. Dendrograms are displayed for both rows and columns. Principal component analysis (PCA) was performed using sklearn PCA algorithm. PCA and hierarchical clustering algorithm is performed on top genes based on mean absolute deviation. StepMiner threshold was used first on the mean absolute deviation numbers to find high values and a second StepMiner threshold was performed to split the high values into top genes based on mean absolute deviation. Differential expression analysis was performed using DESeq2 in R with adjusted pvalue threshold of 0.1 and log2 fold change threshold of 0.5.

**Statistical analyses.** Gene signature is used to classify sample categories and the performance of the multi-class classification is measured by ROC-AUC (receiver operating characteristics area under the curve) values. A color-coded bar plot is combined with a density plot to visualize the gene signature-based classification. All statistical tests were performed using R version 3.2.3 (2015-12-10). Standard t-tests were performed using python scipy.stats.ttest_ind package (version 0.19.0) with Welch's two sample t-test (unpaired, unequal variance (equal_var = False), and unequal sample size) parameters. Multiple hypothesis correction was performed by adjusting p values with statsmodels.stats.multitest.multipletests (fdr_bh: Benjamini/Hochberg principles). The results were independently validated with R statistical software (R version 3.6.1; 2019-07-05). Differential expression analysis was performed in DESeq2 in R. Pathway analysis of gene lists were carried out via the Reactome database and algorithm[90]. Reactome identifies signaling and meta-bolic molecules and organizes their relations into biological pathways and processes. Kaplan–Meier analysis is performed using lifelines python package version

0.14.6. Violin, Swarm and Bubble plots are created using python seaborn package version 0.10.1. Principal component analysis (PCA) was performed using sklearn. The source code for Boolean analysis framework is available at https://github.com/sahoo00/BoNE and https://github.com/sahoo00/Hegemon.

**RNA sequencing**. For polyA capture: Total RNA was assessed for quality using an Agilent Tapestation 4200, and samples with an RNA Integrity Number (RIN) greater than 8.0 were used to generate RNA sequencing libraries using the TruSeq Stranded mRNA Sample Prep Kit with TruSeq Unique Dual Indexes (Illumina, San Diego, CA). Samples were processed following manufacturer's instructions, modifying RNA shear time to five minutes. Resulting libraries were multiplexed and sequenced with 100 basepair (bp) paired end reads (PE100) to a depth of approximately 50 million reads per sample on an Illumina NovaSeq 6000. Samples were demuxltiplexed using bcl2fastq v2.20 Conversion Software (Illumina, San Diego, CA). For ribosomal/globin depletion: Library preparation and sequencing of 30 million 75 or 100 bp paired end reads was conducted using the Illumina's TruSeq RNA Sample Preparation Kit, ribosomal and globin RNA depletion was performed using the Illumina® Ribo-Zero Gold kit and HiSeq 4000 at The Wellcome Center for Human Genetics.

**Human serum cytokines measurement**. Human serum cytokines measurement was performed using the V-PLEX Custom Human Biomarkers from MSD platform. Human serum samples fractionated from peripheral blood of KD and MIS-C patients (all samples collected prior to the initiation of treatments) were analyzed using customized standard multiplex plates as per the manufacturer's instructions.

**Immunohistochemistry**. Formalin-fixed, paraffin-embedded heart tissue sections from COVID19 and KD patients were stained anti-human IL15 receptor A polyclonal antibody (11:200 dilution; proteintech®, Rosemont, IL, USA; catalog# 16744-1-AP) and anti-human IL15 monoclonal antibody (1:10 dilution; Santa Cruz Biotechnology, Inc., Dallas, TX, USA; catalog# sc-8437) after heat-induced antigen retrieval with Tris buffer containing EDTA (pH 9.0). Sections were then incubated with respective HRP-conjugated secondary antibodies followed by DAB and hematoxylin counterstain (Sigma-Aldrich Inc., MO, USA; catalog# MHS1), and visualizing by Leica DM1000 LED (Leica Microsystems, Germany).

**Reporting summary**. Further information on research design is available in the Nature Research Reporting Summary linked to this article.

## Data availability

Source data are provided with this paper. All data is available in the main text or the Supplementary materials. The GEO datasets generated in this work can be accessed at GSE178491. Publicly available datasets used: GSE109351; GSE73464; GSE15297; GSE68004; GSE18606; GSE9863; GSE63881; GSE73577; GSE16797; GSE166489; GSE126124; GSE166489; GSE167028; GSE11545; GSE116946; GSE100150; GSE147608; GSE122552; GSE79970; GSE149050; GSE153781; GSE148810; GSE75023; GSE27864; GSE21835; GSE57253.

## Code availability

The software codes are publicly available at the following links: https://github.com/sahoo00/BoNE[91] and https://github.com/sahoo00/Hegemon[92].

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

## Acknowledgements

This work was supported by the National Institutes for Health (NIH) grants R01-GM138385 (to D.S.), R01-AI141630 (to P.G.), R01DK107585 (to S.D.) and R01-AI155696 (to P.G., D.S., and S.D.), UCOP-RGPO (R01RG3780, R00RG2628 and R00RG2642 to P.G., D.S., and S.D.), PreVAIL R61HD105590, R01HL140898 (to J.C.B. and A.H.T.), and a UC San Diego Stem Cell Center Pilot award (to P.G., D.S., and S.D.). G.D.K. was supported through The American Association of Immunologists Intersect Fellowship Program for Computational Scientists and Immunologists. Clinical sample collection was supported by the Patient Outcomes Research Institute (PCORI) CER-1602–3447 (to J.C.B.), a Gordon and Marilyn Macklin Foundation grant (J.C.B.), and a UC San Diego Stem Cell Center Pilot award (to P.G., D.S., and S.D.). This publication includes data generated at the UC San Diego IGM Genomics Center Utilizing an Illumina NOVASeq 6000 that was purchased with funding from a National Institutes of

Health SIG grant (#S10 OD026929). Figures were created with BioRender.com and assembled using adobe illustrator.

## Author contributions

J.C.B., A.T.H. recruited the KD patients in cohort 1 and conducted the RNA seq studies; D.S., J.C.B. and P.G. conceptualized the project; C.S., A.T.H., Jo.Ka, J.B., J.C.B., Pediatric Emergency Medicine Kawasaki Disease Research Group* recruited the KD and MIS-C patients in cohorts 2 and 3 and collected biological samples used in this study; G.D.K. and C.S. carried out the serum cytokine analysis under the supervision of S.D.; S.K. and G.D.K. carried out the immunohistochemical studies under the supervision of D.S. and P.G.; D.S. carried out all computational modeling and analyses and contributed all software used in this work; Ji.Ki. was responsible for the management of transcriptomic datasets and uploading to the NCBI. D.S., G.D.K., and P.G. prepared figures for data visualization; P.G., G.D.K., C.S. wrote the original draft of the manuscript; D.S., G.D.K., A.H., Jo.Ka., C.S., J.C.B. edited and revised the manuscript. All co-authors approved the final version of the manuscript. D.S., S.D., J.C.B., and P.G. supervised various parts of the project and secured funding; D.S. and P.G. administered the project.

## Competing interests

The authors declare no competing interests.

## Additional information

---

## Pediatric Emergency Medicine Kawasaki Disease Research Group

Naomi Abe[9], Lukas Austin-Page[9], Amy Bryl[10], J. Joelle Donofrio-Ödmann[9,10], Atim Ekpenyong[9,10], Michael Gardiner[9,10], David J. Gutglass[9,10], Margaret B. Nguyen[9,10], Kristy Schwartz[9,10], Stacey Ulrich[9,10], Tatyana Vayngortin[9,10] & Elise Zimmerman[9]

[9]Rady Children's Hospital San Diego, San Diego, CA, USA. [10]Department of Pediatrics, University of California San Diego, San Diego, USA.

