## [Peer Review File · Nature Communications]

REVIEWER COMMENTS

Reviewer #1 (Remarks to the Author):

Review of Sahoo et al submitted manuscript to Nature Communications entitled “An AI-guided invariant signature places MIS-C with Kawasaki disease in a continuum of host immune responses”

In this manuscript, Sahoo et al attempt to delineate some of the similarities and differences between MIS-C and KD using a combination of transcriptional profiling from peripheral whole blood, serum cytokine analysis, and correlative clinical assays and data. They then leverage their RNA data to examine a set of 166 transcripts that they had previously identified as being commonly altered in a variety of infection-induced inflammatory states, including a variety of viral illnesses (including other coronaviruses and influenza, but also non-respiratory viruses such as Hepatitis viruses, Ebola and Zika), as well as bacterial, and fungal infections; this transcript set called a subset of 20 transcripts (called sVIP) whose alterations denote association with severity of disease. Based on their findings, they conclude that MIS-C and KD lie on a continuum of immune responses.

Overall/major issues:

1) While the manuscript is of interest and contains some unique findings, some of the authors' conclusions do not appear to be supported by the data presented. The main issue is that the use of VIP and sVIP are at the crux of the manuscript. Notably the VIP/sVIP gene set is described only as a preprint (thus far over 8 months after release as a bioRxiv article); regardless, the VIP/sVIP gene sets were constructed by using ACE2R as a seed. Despite this, it is not only not specific for SARS-CoV-2 infections, it may not even be specific for viral illnesses, as the 166 transcription signature appears to also be significantly altered in other infections (including bacterial and fungal illnesses), suggesting it is more a non-specific set of transcripts associated with systemic inflammation. In this vein, it would be interesting to know if this gene set was also significantly altered in settings of non-infectious hyperinflammation (e.g. SLE or JIA flares, familial MAS/HLH, or in CART-19 CRS). Further, given that MIS-C is believed to be a post-infectious process, the application of this gene set to MIS-C and KD should be better rationalized and further explained. Overall however, having KD and MIS-C fall along a spectrum comprised predominantly of markers of systemic inflammation may really not be all that surprising, and importantly, the falling of these syndromes within this broad spectrum signature of hyperinflammation does not necessarily indicate a common proximal immune pathogenesis. Indeed, based on the data presented by the authors, it could just as easily be argued that despite some similarities in clinical presentation and certain immune parameters, the fact that IL15 (and MIP1a) is not elevated in KD while it is in MIS-C (data supplied as a supplemental figure that really should be moved to the main figures) may indicate that these two syndromes do not share proximal pathways

of immunopathogenesis, which is important to point out given the importance and central nature of the IL15/IL15R alterations across all conditions in which VIP/sVIP was previously tested.

2) Given that the main point of the manuscript is to delineate similarities or differences between MISC and KD, it is unclear why acute and subacute KD and MISC are not compared to each other, but rather against control groups. In this vein, although the statistics are not provided, Figure 3B and 3C suggest that acute KD and MISC may actually differ significantly from one another. Providing such statistical comparisons would be very important, in order to understand the extent of the similarities and differences between KD and MISC.

3) From a clinical standpoint, the measure of severity in KD that does not present with shock is unresponsiveness to front-line treatments (IVIg), or development of sequelae such as CA disease (and how bad the CA disease is - from mild coronary wall echobrightness to ectasia to frank aneurysms, small and large). It does not appear that the sVIP differentiated KD without CAA from those with small CAA from those with giant CAA, although these statistics are again not calculated/presented. We suggest providing these statistics. If sVIP does not associate with severe KD, then this may further support the issue raised above in #1.

Specific/minor issues:

4) Authors refer to MIS-C as “uncharted territory”; however, many recent papers have contributed to our understanding of the immune pathogenesis. We suggest rephrasing and acknowledging recent work included by Consiglio et al (PMID: 32966765), as well as Ramaswamy et al (PMID: 33891889), and that of others.

5) Results: The authors keep referring to Supplementary Table 1; there is no such table. Instead there is a non-supplemental table (Table 1) followed by a disorganized page of text (on page 32 of merged PDF) which has no title or legend; it is unclear as to why this latter page is provided and what it is meant to show.

6) Results: Figure 2A appears superfluous and the figure in 2B is really a table.

7) Results: In regards to Figure 3, the authors argue that a previously validated 13-transcript signature that differentiates KD from other febrile illnesses was unable to differentiate MIS-C and

KD. The authors use this as evidence that MIS-C and KD exist along the same continuum. However, the authors use a very small number of patients to draw this conclusion. They also do not attempt to compare KD to other inflammatory syndromes in this setting. The authors also speculate that the two scores may have additive diagnostic/prognostic abilities. They do not support this speculation with evidence. We suggest attempting to evaluate the additive nature of these two scores or removing this speculation from the results section and restricting it to future directions in the discussion.

8) Results: In discussing Figure 4, the authors state that “Taken together, these findings suggest that FDA-approved therapeutics targeting TNF α and IL1 β pathways may be beneficial to treat MIS-C.” This has already been recommended by the American College of Rheumatology guidelines (PMID: 33277976) for the treatment of MIS-C. We suggest acknowledging this.

9) Results: In Figure 4D, there are two GSEA for TNF; what is the difference between these? We would suggest explaining this in the results (and perhaps the discussion if relevant), especially as one figure indicates a statistically significant difference between KD AV and MIS-C, while the other does not (we assume there is no statistically significant difference due to lack of p value on the figure), and the authors conclude that “Gene set enrichment analysis on the transcriptomic dataset for each of the differentially expressed cytokines (Figure 4C) showed that the gene sets for those pathways were also induced” in MIS-C at levels significantly higher than KD (Figure 4D-G).

10) Results: The authors discuss the role of eosinophils as a marker of disease severity. However, they cite data only from adult patients; are there any such data in children, since KD and MIS-C are both pediatric conditions. To the best of our knowledge, eosinopenia is not widely used clinically as a marker of disease severity in pediatric patients.

11) Results: Figure 6F and G appear to be superfluous and difficult to draw any true conclusions from, since the data include an n=1 for KD and n=0 for MIS-C; we are not sure that 6F and 6G really shed any appreciable light on the MIS-C/KD similarities and differences question, which is the main point of the manuscript. The authors speculate at the end of the results section about IL15/IL15RA induction in cardiac tissue in MIS-C without any evidence being shown. Overall we recommend removing Figures 6F and 6G and any associated speculation from the results.

12) Methods: The authors should address the timing of blood samples of MIS-C and acute KD patients, with respect to any treatment the patients may have received.

13) Methods: "Subacute" KD should be more prominently defined (it is only in parenthesis in one figure legend and in the collection of blood samples section, which is difficult to find).

14) References: The authors should re-check their citations. In a number of places, reference #12 (Sahoo et al bioRxiv manuscript) is discussed, but reference #12 is actually Tai et al. in the Reference List. Other references are also numbered incorrectly or out of order.

Reviewer #2 (Remarks to the Author):

This manuscript by Sahoo et al. renders insightful notion to unravel the nature of the host immune response (IL15-centric) in MIS-C

and KD. Nevertheless, a couple of issues should be detailed. However, major concern is the title of "...invariant signature places MIS-C with Kawasaki disease in a continuum of host immune responses". They authors show similar of 166-gene viral pandemic (ViP) signature, and its

20-gene severe (s)ViP subset and the difference of 13-transcript signature between MIS-C and KD patients. Besides, many descriptions regarding results require revision.

1. The authors utilized AI-based approaches and a number of datasets to link MIS-C to KD and COVID-19, and showed that MIS-C and KD are in the same host immune continuum, while MIS-C is distinguishable from KD by examining cytokine data. However, the laboratory validation data are not convincing enough to support the conclusion. For instance, the availability of MIS-C specimen could better address the phenotypic issue amid MIS-C and KD.

2. The role of eosinophil in KD has been reported by a series of studies (PMID: 33490002, 33481812, 33363059, 22447381, and 19438983) as well as IL-15 in PMID: 15251134, and 12594117. In this regard, the authors are advised to cite these vital references. In addition, the authors also discuss the their difference with PMID: 33497356 (no increase of IL-15).

3. Figures are required to come clean. For instance, Figure 1A displays a lot of information but not clearly stated in the text. Why are datasets of other species (rodent, swine-flu, bird flu and so on) included? How can these data be of help to support the conclusion? Is there any real data contributing to Figure 1B? If not, Figure 1B is literally out of the blue.

4. Reference 12 reported by Tai et al. has nothing to do with IL15/IL15RA, thereby an incorrect citation and many places of citation of Reference 12 are wrong. As such, the rationale behind IL15/IL15RA as central targets is vague.

5. The authors sought to highlighted IL15-centric response as a vital factor that contributes to MIS-C. However, only observational data is insufficient to gain clinical mechanistic insights into this disease. Thus, the figure 7C is not suitable in the article. The authors do not provide any mechanistic evidence in the figure 7C.

6. We must appreciate the authors about their lots of works and amazing figures.

Reviewer #3 (Remarks to the Author):

In this manuscript, Sahoo et al. employed two gene signatures conserved across all viral pandemics (ViP) to investigate the similarity and difference between Kawasaki Disease (KD) and Multisystem Inflammatory Syndrome in Children (MIS-C). At first, a 166-gene signature was identified from over 45,000 transcriptomic datasets of viral pandemics using a bioinformatics approach termed Boolean Equivalent Correlated Clusters (BECC). This ViP signature and its subset comprised of 20-genes termed severe ViP signature (sViP), capturing the host immune response, are predictive of disease severity in all viral pandemics and outbreaks as well as KD. Based on such prediction results, the authors claimed that ViP signatures are induced in acute KD. Further analyses show that the host immune response determined by the ViP signatures, is similar in KD and MIS-C but is more intensive in MIS-C than KD. Additionally, the similarity between KD and MIS-C was confirmed by the poor performance of classifying KD and MIS-C using a KD specific signature. Finally, the authors demonstrated that the profiling of 10-serum cytokines from one-year convalescent KD samples could differentiate KD and MIS-C. While the analytic procedures are solid and there are some interesting findings, the whole study suffers from the primary use of several small and biased gene signatures and thus the findings are not conclusive. My major concerns are listed below.

1) The aforementioned gene ViP, sViP, and KD signatures are biased towards classification of disease severity but they don't really capture the whole spectrum of true molecular changes in diseases. Dependent on timepoints, normally SARS-CoV-2 can induce changes in hundreds to thousands of genes.

2) Although the 166-gene ViP signature may capture some commonality among different viruses, it doesn't reflect the virus specific changes and again is incomplete.

3) Different tissues have different molecular responses to infectious or environmental stimuli. These signatures were derived from a number of tissues and thus the subsequent analyses lost tissue specificity.

4) Throughout the paper, the authors claimed that the 20-gene sViP signature outperformed the 166-gene ViP signature in predicting disease severity in a number of cohorts but it is unclear which classification method was employed and whether the classifier was retrained for each cohort.

REVIEWER COMMENTS

Reviewer #1 (Remarks to the Author):

Review of Sahoo et al submitted manuscript to Nature Communications entitled “An AI-guided invariant signature places MIS-C with Kawasaki disease in a continuum of host immune responses”

In this manuscript, Sahoo et al attempt to delineate some of the similarities and differences between MISC and KD using a combination of transcriptional profiling from peripheral whole blood, serum cytokine analysis, and correlative clinical assays and data. They then leverage their RNA data to examine a set of 166 transcripts that they had previously identified as being commonly altered in a variety of infection-induced inflammatory states, including a variety of viral illnesses (including other coronaviruses and influenza, but also non-respiratory viruses such as Hepatitis viruses, Ebola and Zika), as well as bacterial, and fungal infections; this transcript set called a subset of 20 transcripts (called sVIP) whose alterations denote association with severity of disease. Based on their findings, they conclude that MISC and KD lie on a continuum of immune responses.

Overall/major issues:

Comment 1) While the manuscript is of interest and contains some unique findings, some of the authors' conclusions do not appear to be supported by the data presented. **The main issue is that the use of VIP and sVIP are at the crux of the manuscript.** Notably the VIP/sVIP gene set is described only as a preprint (thus far over 8 months after release as a bioRxiv article); regardless, the VIP/sVIP gene sets were constructed by using ACE2R as a seed. Despite this, **it is not only not specific for SARS-CoV-2 infections, it may not even be specific for viral illnesses, as the 166 transcription signature appears to also be significantly altered in other infections (including bacterial and fungal illnesses), suggesting it is more a non-specific set of transcripts associated with systemic inflammation.** In this vein, **it would be interesting to know if this gene set was also significantly altered in settings of non-infectious hyperinflammation (e.g. SLE or JIA flares, familial MAS/HLH, or in CART-19 CRS).** Further, given that MISC is believed to be a post-infectious process, the application of this gene set to MISC and KD should be better rationalized and further explained. **Overall however, having KD and MISC fall along a spectrum comprised predominantly of markers of systemic inflammation may really not be all that surprising**, and importantly, the falling of these syndromes within this broad spectrum signature of hyperinflammation does not necessarily indicate a common proximal immune pathogenesis. Indeed, based on the data presented by the authors, it could just as easily be argued that despite some similarities in clinical presentation and certain immune parameters, **the fact that IL15 (and MIP1a) is not elevated in KD while it is in MISC (data supplied as a supplemental figure that really should be moved to the main figures) may indicate that these two syndromes do not share proximal pathways of immunopathogenesis, which is important to point out given the importance and central nature of the IL15/IL15R alterations across all conditions in which VIP/sVIP was previously tested.**

RESPONSE: This is a multi-part comment, which we have broken down into major sub-parts and responded to each:

“...the use of VIP and sVIP are at the crux of the manuscript. Notably the VIP/sVIP gene set is described only as a preprint (thus far over 8 months after release as a bioRxiv article...”- **The reviewer is right that this work largely rests on two signatures that were discovered and reported in another work.** We are pleased to report that the ViP/sViP signatures have now been published in the Lancet Press journal, EBioMedicine. Once published, the article received wide coverage for being the first study that derived a signature(s) from pandemics of the past to prospectively analyze datasets that emerged from the COVID-19 pandemic. Ongoing

work has revealed the importance of these ViP signatures in vetting COVID-19 disease modeling (PMID: 33106807) and these signatures are already helping navigate post-COVID-19 lung and brain disease, providing valuable insights into these new diseases. Although it is too early to assess the impact of these signatures, as independently authored editorial has highlighted just one of those aspects:

INDEPENDENT EDITORIAL: NK cells on the ViP stage of COVID-19. Masselli E, Vitale M. EBioMedicine. 2021 Jul;69:103458. doi: 10.1016/j.ebiom.2021.103458. Epub 2021 Jun 26. PMID: 34186492

“....it is not only not specific for SARS-CoV-2 infections, it may not even be specific for viral illnesses, as the 166 transcription signature appears to also be significantly altered in other infections (including bacterial and fungal illnesses), suggesting it is more a non-specific set of transcripts associated with systemic inflammation.....” - **We agree that this is an important point that we did not preemptively address in the manuscript, and this was a missed opportunity.** This issue of apparent non-specificity to viral infections was explicitly addressed in the original article (excerpt below).

From the now-published EBioMedicine Article: “Notably, the 166-gene host response signature was not specific for viral infections per se; it also performed well in classifying samples with bacterial infections, both in vitro and in vivo, and fungal infections in vivo (**Fig. 3g**). These findings were not surprising because the prominent overrepresentation of interferon signaling that is captured within the signature (**Fig. 3c**) is widely accepted as a shared fundamental aspect of host defense response during any infection [61]. Despite such apparent promiscuity, what is noteworthy is that the ViP signatures were relatively specific for infections/inflammation (**Fig. S3**).” Figure S3 is presented below.

“...., it would be interesting to know if this gene set was also significantly altered in settings of non-infectious hyperinflammation (e.g. SLE or JIA flares, familial MAS/HLH, or in CART-19 CRS)....” - **We agree.** These are certainly very important controls to test, and we thank the reviewer for this suggestion. We have now performed additional analyses on datasets from clinical conditions suggested by the reviewer and other conditions listed below (see **“ACTION(S) TAKEN”** below).

Figure S3. ViP signatures are specific for diseases of infectious and inflammatory conditions. Bubble plots showing up- (red) or downregulation (blue) of ViP signatures in blood samples from patients with diverse conditions and their corresponding controls. The size of bubble indicates the accuracy of classification (AUC ROC; see key) of controls from diseases samples using the signatures. Each dataset contains its own biological controls, and there are no replicates. IBD, Inflammatory bowel disease; NSCLC, non-small cell lung cancer; COPD, chronic obstructive pulmonary disease; Staph, staphylococcal infection; SLE, systemic lupus erythematosus.

“...having KD and MISC fall along a spectrum comprised predominantly of markers of systemic inflammation may really not be all that surprising,..... does not necessarily indicate a common proximal immune pathogenesis.based on the data presented by the authors, it could just as easily be argued that despite some similarities in clinical presentation and certain immune parameters, the fact that IL15 (and MIP1a) is not elevated in KD while it is in MISC (data supplied as a supplemental figure that really should be moved to the main figures) may indicate that these two syndromes do not share proximal pathways of

immunopathogenesis....” – We believe that this criticism rises largely from an unfortunate misinterpretation of the presented data in the manuscript. For the convenience of the reviewer, we have revisited the evidence again below:

- 1) That IL15 is increased in MIS-C but not KD is *not* an accurate conclusion. In fact, throughout the manuscript we have explicitly stated and showcased data that showed that IL15 is the shared cytokine response in both conditions. The degree of elevation is higher in MIS-C than in KD. We showed this finding as direct ELISA-based IL15 assessment studies (S2A in original submission, now in **Fig 3B**); correlation matrix showing that platelets correlate with IL15 in both MIS-C and KD (**Fig 5C**; **S3A-B**). We believe that this unfortunate conclusion that IL15 is elevated in MIS-C and not in KD is might have arisen from us calling out that IL15 correlates with MIP1a (**Fig 5E**) and AEC (**Fig 5F**, bottom panels) in MIS-C, but not KD.
- 2) That MIP1a is increased in MIS-C but not KD is *not* an accurate conclusion. As in the case of IL15, we showed that MIP1a is the shared cytokine response in both conditions. Once again, the degree of elevation is higher in MIS-C than in KD. We showed this finding as direct ELISA-based MIP1a assessment studies (S2A in original submission, now in **Fig 3B**). We believe that this unfortunate misinterpretation stemmed from the fact that we did highlight in the manuscript as a striking difference that MIP1a correlated with PLT (platelet count) in MIS-C, but not KD (**Fig 5D**; **S3A-B**).

It is also possible that this unfortunate misinterpretation could have arisen from the Fig S1 where we showed that the KD-specific 13 gene (KD13) signature has a direct correlation with IL15/IL15RA transcripts in MIS-C, but not KD.

In conclusion, we presented data in main Figures and in Supplementary figures that support the conclusion that both KD and MIS-C share the nature of the hyperinflammation, which resembles that seen in viral pandemics such as in COVID-19, i.e., all of them are predominantly IL15/IL15RA centric.

ACTION(S) TAKEN: In this revised version of the manuscript, we have made the following changes to address the issues:

To ensure that readers do not miss that IL15 is induced in both KD and MIS-C:

- 1) We have now moved the **Supplementary Figure S2A** panel (violin plots showing the relative cytokine levels for IL15, MIP1a etc. in MIS-C vs KD) to main **Figure 4B**. These graphs help visualize the finding that was initially included in the form of a heatmap but lacked the display of p values and data points. We hope that these data make it explicitly clear that IL15 is indeed induced in both KD and MIS-C, just that it is induced at much higher levels in MIS-C, placing it farther along the spectrum, as we had claimed previously.
- 2) We added a new sentence and cited prior work (a recommendation by Reviewer #2) which further highlights that our AI-driven finding of high IL15 in acute KD is consistent with prior work by others. See **Page 7-8**.

spike protein with the host entry receptor, ACE2 is critical for the induction of ViP signatures²². Findings are also in keeping with prior work³⁷ showing that serum levels of IL15 is significantly elevated in acute KD, ~10-fold

7 | Page

compared with subacute-KD and normal controls, and that such increase correlated with the concomitant increase in serum TNF α .

To demonstrate specificity of the ViP signatures amidst the apparent promiscuity:

- 3) **New data to demonstrate signature specificity to infectious conditions (i.e., ability to distinguish them from non-infectious inflammatory diseases):** We analyzed autoimmune inflammatory conditions suggested by this reviewer and additional conditions that are associated with multisystem inflammatory states. We used four signatures to analyze these datasets: ViP, sViP, composite IL15/IL15RA and KD-13 signature. These studies are included in the revised manuscript as a main figure, **Fig 3B**. As seen in the figure below, a variety of autoimmune syndromes showed strong induction of ViP/sViP signatures; these include SLE, JRA, JM, and JIA. Another set of autoimmune syndromes did not; these include JDM, MS, Sarcoidosis, NOMID and MAS that is due to NLRC4 mutation. Furthermore, the signatures were not induced in conditions where the body's immune system is suppressed (e.g., cancers, pregnancy, and post-transplant immune suppressed states, B cell deficiency). These results are not surprising given the fact that many autoimmune diseases (SLE, DM-type 1, JIA, RA, celiac disease and IBD) have now been linked to viruses as trigger points [see Landmark paper: Harley JB, Chen X, Pujato M, et al. Transcription factors operate across disease loci, with EBNA2 implicated in autoimmunity. *Nat Genet*. 2018 May;50(5):699–707]. We also included additional conditions, some of which are known to be infection-coupled inflammation (such as IBD), whereas others are not (pregnancy).

For the convenience of the reviewer, we have placed the text, figures and its legend within this rebuttal document (see below).

Text on Page 9 describing these results:

Because all three conditions represent diseases of the immune system that share an 'infectious trigger', we asked if the ViP/sViP signatures are also induced in the setting of other diseases of the immune system. To this end, we analyzed numerous publicly available datasets, ranging from immunosuppressed states (as negative control), infectious diseases (both viral and bacterial; as positive control), and autoimmune diseases (**Figure 3D**), and assessed the ability of ViP/sViP signatures to classify control and diseased samples in each dataset. Because the ViP/sViP signatures are able to detect the shared core fundamental host immune response in cell/tissue agnostic manner²², we tested diverse samples ranging from whole blood to bronchoalveolar lavage fluid (**Figure 3D**). The ViP/sViP signatures performed as anticipated in the negative and positive control datasets, i.e., neither signature was induced in immunosuppressed conditions, e.g., malignancies, pregnancy, post-transplant immunosuppression, but both were induced in infectious diseases, e.g., sepsis, HIV, RSV, and tuberculosis (*left*; **Figure 3D**). In the case of the autoimmune diseases, the ViP/sViP signatures were induced in some, but not others. The signatures were induced in those conditions that have multifactorial triggers, including potential contributions from infections; for example, mechanistic studies have identified viral link in many of them (EBV-linked autoimmune diseases⁴⁰ such as JIA, SLE, IBD). The signatures were not induced in other conditions where the disease triggers remain mysterious (e.g., sarcoidosis) or where the disease is driven by specific mutations, e.g., Neonatal onset multisystem inflammatory disease (NOMID) that is due to mutant NLRC3 and macrophage activation syndrome (MAS) that is due to mutant NLRC4). These findings lend further support to our finding that ViP/sViP signatures are induced and perform well to identify severe MIS-C, which shares infection as a trigger, much like KD and COVID-19. Intriguingly, numerous infectious and autoimmune diseases shared the IL15/IL15RA-centric cytokine response, which is in keeping with prior observations⁴¹.

Fig 3D: Performance of ViP/sViP signatures on diverse tissues and in diverse diseases of the immune system. Bubble plots of ROC-AUC values (radii of circles are based on the ROC-AUC) demonstrating the strength of classification and the direction of gene regulation (Up, red; Down, blue) in numerous publicly available historic datasets representing diverse diseases of the immune system using 4 gene signatures: the 166-gene ViP signature, the 20-gene sViP signature, the KD-13 signature, and finally the *IL15/IL15RA* composite score. Numbers on top of bubble plots indicate number (n) of control vs. diseased samples in each dataset. Abbreviations: PBMCs, peripheral blood mononuclear cells; Mac, macrophages; WB, whole blood; MTb, *M. tuberculosis*; Flu, Influenza; HIV, Human immunodeficiency virus; RSV, Respiratory syncytial virus; JM, Juvenile myositis; sJIA, Systemic Juvenile Idiopathic Arthritis; SLE, Systemic lupus erythematosus; IBD, Inflammatory bowel disease; COPD, Chronic obstructive pulmonary disease; JDM, Juvenile dermatomyositis; MS, Multiple sclerosis; BAL, Bronchoalveolar lavage; NOMID, Neonatal onset multisystem inflammatory disease; MAS, Macrophage activation syndrome; NLR4, NLR Family CARD Domain Containing 4.

To clarifying what part of the immunopathogenesis appears to be shared and what is distinct between KD and MIS-C:

- 4) On Page 12-13 we have explicitly clarified this. In the first paragraph we addressed what is similar. In the second paragraph we discussed what is different. In the third paragraph we discussed MIS-C specific display immunophenotypes. For the convenience of the editors and the reviewers, we have highlighted all these edits.

Comment 2) Given that the main point of the manuscript is to delineate similarities or differences between MIS-C and KD, it is unclear why acute and subacute KD and MIS-C are not compared to each other, but rather against control groups. In this vein, although the statistics are not provided, Figure 3B and 3C suggest that acute KD and MIS-C may differ significantly from one another. Providing such statistical comparisons would be very important, in order to understand the extent of the similarities and differences between KD and MIS-C.

RESPONSE: We would like to clarify that Figure 3B and 3C provided statistical comparisons of Acute KD and MIS-C with subacute KD. We agree with the reviewer that we missed the comparison between Acute KD

and MISC in Figure 3B and 3C (***)*Because Fig 2 was converted into a Table as per this reviewer's recommendations, Figure 3B-C in the original MS is Figure 2B and 2C in the revised manuscript*(**).

ACTION(S) TAKEN:

- 1) We have added the p-values corresponding to the statistical comparisons in **Figure 2B-C** (ViP signature). P-values of the comparison between Acute KD and MISC is 0.09 for ViP signature and 0.051 for the sViP signature.
- 2) Although not asked by the reviewer, we have also added p-values corresponding to the statistical comparisons in **Figure 2G-H** (KD-13 signature). See below:

pValues	CV	SA	AV	M	FC
SA	0.0441				
AV	1.71e-14	6.43e-05			
M	0.000546	7.21e-05	0.598		
FC	0.42	0.39	2.29e-07	0.000186	

pValues	KD-SA	KD-AV	MISC
KD-AV	8.19e-06		
MISC	7.21e-05	0.621	

We have revised **Figure 2B-C** and **2G-H**, and the corresponding legends and text on **Page #7-8** accordingly.

Because both analyses were not significant, these new findings further provide support to our major claim that KD and MIS-C have shared proximal pathways of host immune response.

Comment 3) From a clinical standpoint, the measure of severity in KD that does not present with shock is unresponsiveness to front-line treatments (IVIg), or development of sequelae such as CA disease (and how bad the CA disease is - from mild coronary wall echobrightness to ectasia to frank aneurysms, small and large). It does not appear that the sViP differentiated KD without CAA from those with small CAA from those with giant CAA, although these statistics are again not calculated/presented. We suggest providing these statistics. If sViP does not associate with severe KD, then this may further support the issue raised above in #1.

RESPONSE: The reviewer raised an important problem, i.e., does the sViP signature distinguish between KD that is complicated by the development of CAA from those that do not.

We performed the direct comparison between CAA- and CAA+S (Small Aneurysm) and CAA+G (Giant Aneurysm) subgroups within the acute KD group. Following are the p values:

P values	CAA- vs CAA+S	CAA- vs CAA+G	CAA+S vs CAA+G
ViP	0.87	0.0013	0.0027
sViP	0.99	0.033	0.074

These new analyses suggest that ViP/sViP signatures do associate with severe KD.

ACTION(S) TAKEN: We have added these KD severity related statistics in **Figure 1F** legend and discussed results in the main text on **Page #6**. For the convenience of the reviewer, we have copied and pasted the text below and highlighted the inclusion of the new analyses:

Finally, we tested the association between sViP signatures and markers of disease severity. Because CAA diameter is a predictor of coronary sequelae (thrombosis, stenosis, and obstruction)^{24, 25} and subsequent major adverse cardiac events (unstable angina, myocardial infarction, and death²⁶), we used the development of coronary artery aneurysms (CAA) as a marker of disease severity. We found that both ViP signatures differentiated acute KD with giant CAAs (defined as a z-score of ≥ 10 or a diameter of ≥ 8 mm^{27, 28}) from convalescent KD samples (ROC AUC 0.95 and 0.97 for ViP/sViP signatures, respectively; **Figure 1F**). The ViP signature effectively classified KD patients with giant aneurysms (CAA-giant) from those with either no aneurysms (CAA-; p value 0.0027) or small aneurysms (CAA-small; p value p value 0.0013). Similarly, the sViP signature effectively classified KD patients with giant aneurysms (CAA-giant) from those with no aneurysms (CAA-; p value 0.033). Such an analysis was not possible in Cohort 2 (**Supplementary Information 1**) because of the smaller cohort size and absence of subjects with giant CAAs.

Although not asked by this reviewer, in the case of MIS-C, we could not perform such an analysis on our cohort (due to insufficient number of subjects with myocardial dysfunction). However, we found the same trend in two new MISC datasets that were annotated with disease severity (GSE166489, Moderate vs Severe, p value = 0.011; and GSE167028, MIS-C +/- myocardial dysfunction, p value = 0.08). This new dataset/analysis is added in **Fig 3A** of this revised submission and the text has been updated on **Page 8-9**. For the convenience of the reviewer, we have copied and pasted the text below:

The sViP signature can recognize severe form of MIS-C that presents with myocardial dysfunction

Next, we asked if the sViP signature can track disease severity in MIS-C. Because of the limited number of 'severe' cases in our MIS-C cohort, we prospectively analyzed two recently accessible MIS-C cohorts (GSE166489¹⁵ and GSE167028³⁹). While both datasets analyzed PBMCs from MIS-C subjects, and both studies used the presence of myocardial dysfunction as basis for severe disease, each study used a slightly different criterion for classification of disease severity (**Figure 3A**). de Cevins et al.,³⁹ classified MIS-C as severe when the patients presented with elevated cardiac troponin I and/or altered ventricular contractility by echocardiography, and clinical signs of heart failure requiring ICU support. Ramaswamy et al.,¹⁵ classified MIS-C as severe if they were critically ill, with cardiac and/or pulmonary failure. In both cohorts, sViP was able to classify severe MIS-C (with myocardial dysfunction; MYO+) from mild-moderate disease (who recovered or presented without myocardial dysfunction; MYO-) (**Figure 3B-C**); while the p value was significant in GSE166489 (**Figure 3B**), a similar trend was conserved in GSE167028³⁹ (**Figure 3C**). These findings show that the sViP

8 | Page

signature can identify severe MIS-C who are at risk to develop myocardial dysfunction, just as it did in the case of KD subjects who are at risk of developing giant CAAs (**Figure 1F**) and similar to its prior performance in identifying adults with COVID-19 who are at risk of respiratory failure, mechanical ventilation, prolonged hospitalization and/or death²².

Taken together with the prior findings, we conclude that the 20-gene sViP signature captures a core set of genes that are expressed in the setting of an overzealous (prolonged or intense, or both) host immune response in all three diseases— KD, MIS-C (this work) and COVID-19²² – despite the fact that each present with distinct clinical features of severity.

Figure 3: Performance of ViP/sViP signatures on independent MIS-C datasets and on diverse tissues and in diverse diseases of the immune system. A-C. Severe (s)ViP signature can classify severe MIS-C based on transcriptomic analysis of blood that was prospectively collected at admission in two independent studies (GSE166489¹⁵ and GSE167028³⁹). Schematic in **A** summarizes the definition of severe MIS-C used by two independent studies. **B-C.** Bar (top) and violin (bottom) plots display the classification of blood samples in two cohorts of MIS-C subjects, based on the need for ICU management due to the presence (MYO+) or recovery in the absence (R or MYO-) of myocardial dysfunction using sViP signature. Welch's two sample unpaired t-test is performed on the composite gene signature score to compute the p values. **D.** Bubble plots of ROC-AUC values

Specific/minor issues:

Comment 4) Authors refer to MIS-C as “uncharted territory”; however, many recent papers have contributed to our understanding of the immune pathogenesis. We suggest rephrasing and acknowledging recent work included by Consiglio et al (PMID: 32966765), as well as Ramaswamy et al (PMID: 33891889), and that of others.

Specific/minor issues:

Comment 4) Authors refer to MIS-C as “uncharted territory”; however, many recent papers have contributed to our understanding of the immune pathogenesis. We suggest rephrasing and acknowledging recent work included by Consiglio et al (PMID: 32966765), as well as Ramaswamy et al (PMID: 33891889), and that of others.

RESPONSE: Agree. That was a poor choice of wording. It is true that many other impactful studies have been published that we should duly acknowledge.

ACTION(S) TAKEN: Rephrasing done along the lines suggested by this reviewer (on **Page 4**). We have now expanded the introduction section to dedicate an entire paragraph on **Page 3** to highlight prior work. For the convenience of the reviewer, we have copied and pasted the section below.

As for the immunopathogenesis of MIS-C, limited but key insights have emerged rapidly, most of which focus on the differences between MIS-C and KD. For example, Gruber et al.,¹² and Consiglio et al.,¹³ showed that the inflammatory response in MIS-C shares several features with KD, but also differs from this condition with respect to T cell subsets¹³. These conclusions were generally supported by two other studies, by Vella et al.,¹⁴ and Ramaswamy et al.¹⁵ who also showed that severe MIS-C patients displayed skewed memory T cell TCR repertoires and autoimmunity characterized by endothelium-reactive IgG. Finally, Carter et al.,¹⁶ reported activation of CD4⁺CCR7⁺ T cells and $\gamma\delta$ T cell subsets in MIS-C, which had not been reported in KD, which made them conclude that MIS-C may be a distinct immunopathogenic illness. While these studies further our understanding of MIS-C and the major conclusions of these studies are comprehensively reviewed elsewhere¹⁷, it is noteworthy that each of these studies had some notable limitations— (i) in Gruber et al.,¹² most of the MIS-C subjects were on immunomodulatory medications when samples were drawn; (ii) in Vella et al.,¹⁴ absence of contemporaneously analyzed healthy pediatric samples which were not available during the early phase of the pandemic; (iii) in Carter et al.,¹⁶ KD subjects were not concurrently studied and the authors themselves acknowledged that such side-by-side immunophenotyping of MIS-C and KD would be necessary to draw

3 | Page

conclusions convincingly regarding similarities and differences between these two syndromes; (iv) absence of validation studies in independent cohorts in them all.

Comment 5) Results: The authors keep referring to Supplementary Table 1; there is no such table. Instead there is a non-supplemental table (Table 1) followed by a disorganized page of text (on page 32 of merged PDF) which has no title or legend; it is unclear as to why this latter page is provided and what it is meant to show.

RESPONSE: It appears that when Excel tables are uploaded, they lose formatting in PDFs created by the journal's upload portal. It is also entirely possible that we created this confusion, inadvertently, when we included Supplementary Tables, we mistakenly labeled them as "Table". We apologize for putting the reviewer(s) through this extra barrier in assessing the manuscript.

ACTION(S) TAKEN: In this revised submission, we have followed Nat Comm guidelines and this reviewer's suggestion and taken the following steps:

- Only 1 Table in main Article (As recommended by this reviewer, this is basically Fig 2B).
- Two Supplementary Tables, which, as per Nat Comm Editorial guidelines, we have uploaded as part of Supplementary Online Materials as:
 - o **Supplementary Information 1** and
 - o **Supplementary Information 2**

Comment 6) Results: Figure 2A appears superfluous and the figure in 2B is really a table.

RESPONSE: Agree.

ACTION(S) TAKEN: We removed Fig 2A and converted Table in Fig 2 to Table 1. The remaining figure numbering was updated accordingly and citations in text were adjusted to reflect these changes.

Comment 7) Results: In regards to Figure 3, the authors argue that a previously validated 13-transcript signature that differentiates KD from other febrile illnesses was unable to differentiate MIS-C and KD. The authors use this as evidence that MIS-C and KD exist along the same continuum. However, the authors use a very small number of patients to draw this conclusion. They also do not attempt to compare KD to other inflammatory syndromes in this setting. The authors also speculate that the two scores may have additive diagnostic/prognostic abilities. They do not support this speculation with evidence. We suggest attempting to evaluate the additive nature of these two scores or removing this speculation from the results section and restricting it to future directions in the discussion.

RESPONSE: This issue--of whether the two signatures sViP and KD-13 synergize-- is an important point raised by the reviewer, one that we alluded to in the original manuscript, but failed to provide any evidence.

In doing so, our *sole intention* was to inform the reader that KD-13 and the ViP signatures had a completely different set of genes and hence, perhaps captured unrelated gene regulatory events. The evidence we provided to support that statement was **Supplementary fig. S1** in which we demonstrated the lack of any form of correlation between the two signatures. Based on this, we speculated that perhaps these signatures work independent of each other and a refined combination of both may have additive effect.

During this revised submission, we have now **carried out additional analyses** to put that speculation to test. We have since determined that **a combination of KD13 and sViP signatures** was able to distinguish both KD from other febrile illnesses and MIS-C in males. We used following equation for the combination $KD13 + 0.71 * sViP$ and achieve borderline significance (p value = 0.05) for both. Further research into this may provide additional benefits by selecting a subset of genes from both signatures.

ACTION(S) TAKEN: In the absence of more evidence beyond what we have at present, we have now removed the sentence where the speculation was raised. For the convenience of the reviewer, we have copied and pasted the new passage below (**Page 8**).

between MIS-C and **KD in either cohort**. Furthermore, a correlation test demonstrated that the two non-overlapping signatures, **sViP** and KD-13, both of which are significantly induced in **KD** and MIS-C (**Figure 2C**) are independent of each other (**Supplementary fig. S1**). **This suggests that these two signatures reflect two fundamentally distinct and unrelated biological domains within the host immune response; whether their diagnostic/prognostic abilities may have an additive benefit remains to be explored.**

Comment 8) Results: In discussing Figure 4, the authors state that “Taken together, these findings suggest that FDA-approved therapeutics targeting TNF α and IL1 β pathways may be beneficial to treat MIS-C.” This has already been recommended by the American College of Rheumatology guidelines (PMID: 33277976) for the treatment of MIS-C. We suggest acknowledging this.

RESPONSE: We thank the reviewer for pointing this out. This paper was published after our work was concluded, and hence, is an important citation that only speaks to the strength of the conclusions drawn from our study.

ACTION(S) TAKEN: We have now done this on Page 12 of this revised submission.

Comment 9) Results: In Figure 4D, there are two GSEA for TNF; what is the difference between these? We would suggest explaining this in the results (and perhaps the discussion if relevant), especially as one figure indicates a statistically significant difference between KD AV and MISC, while the other does not (we assume there is no statistically significant difference due to lack of p value on the figure), and the authors conclude that “Gene set enrichment analysis on the transcriptomic dataset for each of the differentially expressed cytokines (Figure 4C) showed that the gene sets for those pathways were also induced” in MIS-C at levels significantly higher than KD (Figure 4D-G).

RESPONSE: We thank the reviewer for raising these points about the GSEA-related graphs. In the original version of the manuscript, we had used the gene signatures (at MSigDB) just like we used the ViP/sViP/KD-13 signatures—i.e., a composite score of their relative expression in the datasets. In that sense, it was not the typical GSEA analysis.

ACTION(S) TAKEN:

We have now replaced the violin plots with GSEA analysis of the pathways initially derived from the cytokine analyses. These new analyses are presented in panels **Figure 4D-H** (replacing the unconventional use of gene sets as violin plots of composite scores in the original manuscript). For the convenience of the reviewer, we have placed the Figure panels and the corresponding legend below:

Figure 4 D-F. Gene Set Enrichment Analysis (GSEA pre-ranked analysis) of three pathways derived from MSigDB: SANA TNF SIGNALING UP, TIAN TNF SIGNALING VIA NFkB, and SANA RESPONSE TO IFNG UP demonstrate the significance of TNF and IFNG pathway activation in MIS-C. **G-H.** Down-regulated genes after IL1B and IL10 stimulation were derived

from differential expression analysis of GSE44722 (n = 269 genes) , and GSE61298 (n = 208 genes) respectively. GSEA pre-ranked analysis to test the significance of IL1B and IL10 pathway is performed like panel D-F using the down-regulated genes. Source data are provided as a Source Data file.

Comment 10) Results: The authors discuss the role of eosinophils as a marker of disease severity. However, they cite data only from adult patients; are there any such data in children, since KD and MIS-C are both pediatric conditions. To the best of our knowledge, eosinopenia is not widely used clinically as a marker of disease severity in pediatric patients.

RESPONSE: We appreciate this question from the reviewer and recognize that we had unfortunately (and unintentionally) cited only adult literature and left out the appropriate citations of pediatric literature on eosinopenia. We have rectified that now.

ACTION(S) TAKEN: The following references were added on **Pages 11 and discussed on Page 13** of the revised submission to provide the readers with sufficient insights into eosinopenia in both children and adults:

- 1) Pediatric covid-19 is associated with eosinopenia (29.5%)¹.
- 2) Eosinopenia in children is a prognostic factor for ICU admissions². Eosinopenia, defined as an eosinophil count < 15 cells/ μ L and an eosinophil percentage < 0.25%, (hazard ratio [HR]: 2.96; P = 0.008) along with a Pediatric Index of Mortality (PIM) 2 (HR: 1.03; P = 0.004) were both determined to be independent predictors of mortality in the PICU.
- 3) ~50% patients with COVID-19 between ages 20-80 had eosinopenia^{3,4}.
- 4) ~80% patients who died of COVID-19 had eosinopenia⁵.
- 5) Eosinophil levels improved in all patients before discharge⁶, suggesting that resolution of eosinopenia may be an indicator of improving clinical status. Independent studies have verified this finding^{7,8}.

For the convenience of the Editor and the Reviewers we have copied and pasted the section below from Page 11 (edits = yellow):

defense⁶¹⁻⁶³. Persistent thrombocytopenia carried higher mortality in sepsis^{64, 65}, and in COVID-19^{66, 67}. Our analysis revealed a direct and unusually strong correlation between thrombocytopenia and eosinopenia in MIS-C. Eosinophils, on the other hand, as reviewed elsewhere⁶⁸, have important antiviral properties, attributed to their granular proteins (e.g., eosinophil-derived neurotoxin, cationic protein) that display antiviral activities against single-stranded RNA viruses. Eosinophils can also support viral clearance⁶⁹. Eosinopenia, in the setting of acute infection, has been found to be a direct response to infectious stimuli⁷⁰, TLR4 ligands and chemotactic factors⁷¹, and has been considered a reliable diagnostic marker of infection⁷² in critically ill patients and a predictor of mortality⁷²⁻⁷⁴. Of relevance to the pediatric syndrome MIS-C, eosinopenia is encountered in about a 1/3rd of the pediatric COVID-19 subjects⁷⁵. It is noteworthy that eosinopenia (defined as an eosinophil count < 15 cells/ μ L and an eosinophil percentage < 0.25%) is a known poor prognostic factor for admissions into the pediatric ICU (hazard ratio [HR]: 2.96; P = 0.008⁷⁶). It is possible that the two related clinical/laboratory parameters (low PLT and AEC) may be useful indicators of disease severity and prognosis in MIS-C and may guide decision-making in therapy and level of care in the hospital setting.

Comment 11) Results: Figure 6F and G appear to be superfluous and difficult to draw any true conclusions from, since the data include an n=1 for KD and and n=0 for MISC; we are not sure that 6F and 6G really shed any appreciable light on the MISC/KD similarities and differences question, which is the main point of the

manuscript. The authors speculate at the end of the results section about IL15/IL15RA induction in cardiac tissue in MIS-C without any evidence being shown. Overall we recommend removing Figures 6F and 6G and any associated speculation from the results.

RESPONSE: We appreciate this suggestion from the reviewer. We have weighed the pros and cons of eliminating this Figure and see merit on both sides of the argument. Ultimately, we decided that there is a need to highlight the divergent cardiac phenotypes that define severity of KD and MIS-C. Our rationale to include these were further solidified during the course of the revisions—especially when we strived to answer the questions that were posed by this reviewer--- which led us to now strengthen the link between sViP and CAA in KD and sViP and MIS-C (see Response to Comment #3 above). While we acknowledge that $n = 1$ and $n = 0$ is not credible (beyond the purposes of case reports), that IL15 is induced in the heart is not the main message of the paper. Instead, it is our attempt to connect hyperinflammation and one of the target organs. Given that mortality in KD and MIS-C is rare and invasive cardiac procedures are not performed in these settings for research purposes, we have limited access to this type of sample. We have removed speculation about these results as requested by the reviewer.

Comment 12) Methods: The authors should address the timing of blood samples of MIS-C and acute KD patients, with respect to any treatment the patients may have received.

RESPONSE: This information was included in the form of an explicit statement that was in Methods (under a subtitle: “Collection of blood samples and RNA isolation”). We apologize that this was not easily accessible and yet a very important detail. Pre-treatment sample collection is an important aspect of our study because some of the initial published MIS-C immunophenotyping work was done on almost all samples collected after the initiation of treatment (Gruber et al.,⁹).

ACTION(S) TAKEN:

- 1) We have now moved the methods section to the main text (see **Page #14**).

Collection of blood samples and RNA isolation: Whole blood was collected into PAXgene® tubes (PreAnalytiX) for RNA and into red top tubes for serum before the initiation of any treatment (illness day ≤ 10) for the KD, MIS-C and FC subjects and at the clinic visit (day 17-25 of illness for subacute and day 28-320 of illness for late convalescent) for the KD subjects. RNA was extracted following manufacturer’s instruction (PAXgene Blood miRNA Kit). Serum was separated immediately by centrifugation and stored at -80°C until use.

- 2) Timing of blood draw has been explicitly stated also in the results section on **Page #7**, and again in Methods section on **Page #16**.

Comment 13) Methods: “Subacute” KD should be more prominently defined (it is only in parenthesis in one figure legend and in the collection of blood samples section, which is difficult to find).

RESPONSE: We agree that such important information may be difficult to access when present in small font in figure legend. In the original version of the manuscript, this information was also included in the methods section (under a subtitle: “Collection of blood samples and RNA isolation”). However, this information was also not readily accessible because it appeared in the Supplementary File.

ACTION(S) TAKEN: In this revised version of the manuscript, we have now included the information in a few more places:

- 1) The results section clearly defines acute, subacute and late convalescent stages of disease when these terms are used in the manuscript for the first time on **Page 6**. For the convenience of the reviewer, we have placed the edited paragraph below, with highlighted edits.

We next confirmed that both the ViP signatures are induced in acute KD (at presentation, < 10 day of illness) compared to convalescent KD (day 28-320 of illness) in a large new cohort of consecutive patients (n=105) who were diagnosed with the disease prior to the onset of the COVID-19 pandemic (Cohort 1; **Supplementary Information 1**) (**Figure 1D**). Again, the sViP signature outperformed the ViP signature in sample classification (ROC AUC 0.91 vs. 0.74). In an independent cohort (Cohort 2, n=20, **Supplementary Information 1; Figure 1E**) prospectively enrolled in the current study after the onset of the COVID-19 pandemic, the ViP signatures could differentiate the acute from subacute (~10-14 d after discharge; ~day 17-25 of illness) KD samples. As before, the 20-gene sViP signature outperformed the 166-gene ViP signature.

- 2) The *Methods* section has been moved to the main text as per *Nature Comm* formatting and guidelines. Hence, the definition of subacute is now on **Page #14**.

Comment 14) References: The authors should re-check their citations. In a number of places, reference #12 (Sahoo et al bioRxiv manuscript) is discussed, but reference #12 is actually Tai et al. in the Reference List. Other references are also numbered incorrectly or out of order.

RESPONSE: We thank the reviewer for catching this glitch in ENDNOTE.

ACTION(S) TAKEN: We have fixed this now.

Reviewer #2 (Remarks to the Author):

This manuscript by Sahoo et al. renders insightful notion to unravel the nature of the host immune response (IL15-centric) in MIS-C and KD. Nevertheless, a couple of issues should be detailed. However, major concern is the title of "...invariant signature places MIS-C with Kawasaki disease in a continuum of host immune responses". They authors show similar of 166-gene viral pandemic (ViP) signature, and its 20-gene severe (s)ViP subset and the difference of 13-transcript signature between MIS-C and KD patients. Besides, many descriptions regarding results require revision.

RESPONSE TO GENERAL COMMENTS: We are pleased to see that this reviewer found our manuscript as one that renders insights into the nature of the host immune response in MIS-C and KD. He/she, however, found a couple of issues. One of those was the title that we used. The reviewer also felt that the results were inadequately discussed.

ACTION(S) TAKEN:

Title: Based on the reviewer's critiques, we were unsure which specific part of the title was inappropriate. In the absence of such clues, or any suggestions, we interpret this comment as the title might have appeared vague or that some terminologies ("continuum") might be a jargon that is inaccessible to many readers. After taking these thoughts into consideration, we have now changed the title to: "*An AI-guided signature reveals the nature of the shared proximal pathways of host immune response in MIS-C and Kawasaki disease*". We hope

that the reviewer will find this title appropriate in light of the revisions that were made to home in on this central conclusion.

Abstract: We have also edited the abstract to draw contrasts between what are shared (convergent) and what are some of the distinct (divergent) features of KD and MIS-C.

Comment 1. The authors utilized AI-based approaches and a number of datasets to link MIS-C to KD and COVID-19, and showed that MIS-C and KD are in the same host immune continuum, while MIS-C is distinguishable from KD by examining cytokine data. However, the laboratory validation data are not convincing enough to support the conclusion. For instance, the availability of MIS-C specimen could better address the phenotypic issue amid MIS-C and KD.

RESPONSE- Comment 1: This comment brings up the relatively small number of MIS-C subjects in our analysis. We agree. Although most published molecular work on MIS-C subjects have used similar small cohorts, ranging from ~7-25, many of which are in high profile journals (*Cell*⁹, *Science Immunology*¹⁰, and *Nature Medicine*¹¹), our study has distinct advantages over those studies. However, we had not done a good job of highlighting these. We had also failed to explicitly address the limitations of our study.

ACTION(S) TAKEN: In the revised version of the manuscript, we have taken the following measures to tackle the deficiencies raised in this comment.

We have added a section in **Introduction** section (on **Page #3-4**) to highlight limitations of the prior studies that were uniquely overcome in the current study:

them conclude that MIS-C may be a distinct immunopathogenic illness. While these studies further our understanding of MIS-C and the major conclusions of these studies are comprehensively reviewed elsewhere¹⁷, it is noteworthy that each of these studies had some notable limitations— (i) in Gruber et al.,¹² most of the MIS-C subjects were on immunomodulatory medications when samples were drawn; (ii) in Vella et al.,¹⁴ absence of contemporaneously analyzed healthy pediatric samples which were not available during the early phase of the pandemic; (iii) in Carter et al.,¹⁶ KD subjects were not concurrently studied and the authors themselves acknowledged that such side-by-side immunophenotyping of MIS-C and KD would be necessary to draw

3 | Page

conclusions convincingly regarding similarities and differences between these two syndromes; (iv) absence of validation studies in independent cohorts in them all.

Validation in an independent MIS-C cohort: We have analyzed an additional MIS-C dataset (GSE166849; an independent cohort that was publicly released while our work was in review). The findings of this analysis have now been added in the revised manuscript (**Figure 3A-C**). [See **Page #8** of Rebuttal]

Study Limitations: We have explicitly acknowledged the limited access to MIS-C samples in the **Discussion** section on **Page 13**.

The strength of our study lies in the concurrent analysis of KD and MIS-C samples, our access to relatively large and independent cohorts of patients (in the case of KD), our ability to include age-matched pediatric healthy controls and febrile controls (non-KD and non-MIS-C, both pre-pandemic), and that all samples were drawn prior to the initiation of treatment. In doing so, this study overcomes some of the limitations of prior studies^{12, 14, 16}. Another strength is the use of a set of ViP signatures (that were validated in COVID-19)²² and a KD-diagnostic signature³⁸ as the computational framework to compare the two syndromes. Last, but not the least, the multi-omics approach used here on samples obtained from the same patients allowed us to predict and validate the prominent and invariant upregulation of one cytokine pathway (i.e., *IL15*) at both transcript and protein level. Notable limitations of our study include a relatively small sample size of MIS-C subjects (n = 12), limited number of publicly available MIS-C datasets for independent validation, and our inability to access cardiac tissue from KD and MIS-C subjects. Future studies on emerging datasets will enable rigorous validation of the analysis presented here.

Comment 2. The role of eosinophil in KD has been reported by a series of studies (PMID: 33490002, 33481812, 33363059, 22447381, and 19438983) as well as IL-15 in PMID: 12594117, and 12594117. In this regard, the authors are advised to cite these vital references. In addition, the authors also discuss the their difference with PMID: 33497356 (no increase of IL-15).

RESPONSE- Comment 2: We thank the reviewer for this helpful suggestion.

ACTION(S) TAKEN:

- All references for eosinophilia in KD (PMID: 33490002, 33481812, 33363059, 22447381, and 19438983) have been added on **Page #11**
- The reference for IL15 in KD (PMID: 12594117) has now been included on **Page #7**.
- The reference for IL15 in autoimmune diseases (PMID: 12594117) has now been included on **Page #9**

Comment 3. Figures are required to come clean. For instance, Figure 1A displays a lot of information but not clearly stated in the text. Why are datasets of other species (rodent, swine-flu, bird flu and so on) included? How can these data be of help to support the conclusion? Is there any real data contributing to Figure 1B? If not, Figure 1B is literally out of the blue.

RESPONSE- Comment 3: These questions pertain to the very first paragraph in results. In this paragraph, we were introducing the computational framework discovered in our prior work (i.e., the ViP signatures, identified in Sahoo D. et al., EBioMedicine. 2021. PMID: 34127431) that we used for analyzing KD and MIS-C datasets in the present work.

Figure 1A-B were intended to provide a pictorial summary of the prior work, touching upon 5 key aspects:

- A- methodology used to identify the ViP/sViP signatures (i.e., BECC);
- A- the computational rigor (# of datasets and their diversity);
- A- the major evidence-based claims and translational impact; and
- B- the ability of ViP signatures to broadly capture a fundamental infection-inflammation related host response to diverse classes of pathogens.

Because the ViP signatures are the premise of this entire study, we felt that pictorial summaries will help inform the readers how they were derived and what do they mean. This was especially important because the ViP signatures were still unpublished (on Preprint) and are not widely-used entities and lack name recognition. In

retrospect, we realize that although the Figure panels were well-intended, the corresponding text was not very clear, as written, and the line between what was done previously and what is new remained unclear.

Such lack of clarity could have unfortunately led the reviewer to misinterpret Figure 1A as if we have used other species (rodent, swine-flu, bird flu, etc) in the current work on KD/MIS-C, and Figure 1B appeared to be out of the blue.

ACTION(S) TAKEN: In this revised submission, we have **edited the text** extensively to achieve this clarity by annotating the Figure legends (**1A-B**) and the corresponding text. We have also attempted to better explain the ViP signatures (prior work) on **Page 4-5**.

Comment 4. Reference 12 reported by Tai et al. has nothing to do with IL15/IL15RA, thereby an incorrect citation and many places of citation of Reference 12 are wrong. As such, the rationale behind IL15/IL15RA as central targets is vague.

RESPONSE- Comment 4: There was a glitch in the ENDNOTE traveling library which caused this unfortunate error. Reference #12 was meant to be Sahoo D. et al., EBioMedicine. 2021 Jun;68:103390. doi: 10.1016/j.ebiom.2021.103390. Epub 2021 Jun 11. PMID: 34127431].

ACTION(S) TAKEN: We have fixed this citation error in this revised submission.

Comment 5. The authors sought to highlighted IL15-centric response as a vital factor that contributes to MIS-C. However, only observational data is insufficient to gain clinical mechanistic insights into this disease. Thus, the figure 7C is not suitable in the article. The authors do not provide any mechanistic evidence in the figure 7C.

RESPONSE- Comment 5: We agree.

ACTION(S) TAKEN: In this revised version of the manuscript, we have now removed **Figure 7C**, its citation in text and the corresponding legend and only discussed the possible implications of the findings/evidence that we have presented in this work.

Comment 6. We must appreciate the authors about their lots of works and amazing figures.

RESPONSE- Comment 6: We thank the reviewer for acknowledging the time and effort that went into creating the display items.

ACTION(S) TAKEN: None required.

Reviewer #3 (Remarks to the Author):

In this manuscript, Sahoo et al. employed two gene signatures conserved across all viral pandemics (ViP) to investigate the similarity and difference between Kawasaki Disease (KD) and Multisystem Inflammatory Syndrome in Children (MIS-C). At first, a 166-gene signature was identified from over 45,000 transcriptomic datasets of viral pandemics using a bioinformatics approach termed Boolean Equivalent Correlated Clusters (BECC). This ViP signature and its subset comprised of 20-genes termed severe ViP signature (sViP), capturing the host immune response, are predictive of disease severity in all viral pandemics and outbreaks as well as KD. Based on such prediction results, the authors claimed that ViP signatures are induced in acute KD. Further analyses show that the host immune response determined by the ViP signatures, is similar in KD and MIS-C but is more intensive in MIS-C than KD. Additionally, the similarity between KD and MIS-C was confirmed by the poor performance of classifying KD and MIS-C using a KD specific signature. Finally, the authors

demonstrated that the profiling of 10-serum cytokines from one-year convalescent KD samples could differentiate KD and MIS-C. While the analytic procedures are solid and there are some interesting findings, the whole study suffers from the primary use of several small and biased gene signatures and thus the findings are not conclusive. My major concerns are listed below.

RESPONSE TO GENERAL COMMENTS: We are pleased to see that this reviewer found that our “*analytic procedures are solid and there are some interesting findings*”, but there are major criticisms that he/she points to, all of which pertain to the credibility of the major tool we used here, i.e., the ViP/sViP signatures. We agree that if the premise of the study is not clearly presented, this is something that needs to be fixed. The way the ViP signatures were derived (how was this model trained, and that it lacks pathogen or tissue specificity) is also a key aspect that the reviewer brings up as good questions, which we had not explained clearly to inform the reader. We have now tried to address these issues through our point-by-point responses below and exhaustive edits throughout the manuscript. We have also added additional data (as requested by **Reviewer #1**) that highlight the specificity of ViP signature amidst the apparent promiscuity of its ability to classify almost all viral infections and diverse samples.

Comment 1) The aforementioned gene ViP, sViP, and KD signatures are **biased towards classification of disease severity** but they **don’t really capture the whole spectrum of true molecular changes in diseases**. Dependent on timepoints, normally SARS-CoV-2 can induce changes in hundreds to thousands of genes.

RESPONSE- Comment 1: This comment refers to the three signatures -- ViP, sViP, and KD-13 –that we used here to assess the similarities and differences between MIS-C and KD. We are not sure why the reviewer concludes that these signatures are ‘biased’ towards classification of disease severity and what he/she implies by ‘whole spectrum of true molecular changes’. It is entirely possible that this unfortunate misconception arose because the original manuscript describing ViP signatures was not yet published at that time and not explained in detail. It is also possible that we did a poor job of clarifying what these signatures meant and how they were derived.

- **ViP signature (166-gene)** is comprised of a set of genes that are invariably upregulated in any viral pandemic in acute stage and are subsequently invariably downregulated during convalescence. The reviewer is right that thousands of genes are induced during a viral infection, but the computational tool used here, called Boolean Equivalent Correlated Clusters (BECC) algorithm only focuses on Boolean equivalent relationships to identify potentially functionally related gene sets. In doing so, this approach is geared to exclusively focus on those that are part of the invariant spectrum of the host response, while deprioritizing the genes that are part of the variable spectrum of the host response. The resultant ViP signature is induced in all datasets derived from viral infection, and by that token, *represent a fundamental aspect of the host response to viral infections*. In the original MS describing this signature (now published; Sahoo D et al., EBioMedicine, 2021), we presented evidence for the ability of the ViP signature to classify infected from uninfected control samples across a variety of pandemics, and hence, *the ViP signature was not biased to identify severity, but biased to identify the invariant host response shared across viral infections*.
- **sViP signature (20-gene)** is a subset of the ViP signature that is biased to identify disease severity. It was derived by rank-ordering the 166-genes in ViP using the largest available disease severity-annotated dataset. *The sViP signature was biased to identify severity*
- **KD signature (13-gene)** was developed¹² purely as a **diagnostic tool** that is supposed to distinguish KD from non-KD febrile illnesses of all other kinds. Based on what was published and how this signature was derived, *this signature was not biased to detect disease severity*.

ACTION(S) TAKEN: We have expanded the **introduction section (Page #4)** and **Results section (Page #5 and 7)** to include pertinent details on **how the ViP signature was derived and how it was trained**. Some of these edits also address the questions below.

Comment 2) Although the 166-gene ViP signature may capture some commonality among different viruses, it doesn't reflect the virus specific changes and again is incomplete.

Comment 3) Different tissues have different molecular responses to infectious or environmental stimuli. These signatures were derived from a number of tissues and thus the subsequent analyses lost tissue specificity.

RESPONSE- Comments 2-3: We had not intended to identify virus or tissue specific gene signatures; the computational methodology used to identify the ViP signatures was specifically chosen for its ability to identify the fundamental invariant shared host response pattern to viral pandemics, while ignoring the virus/host/tissue specific differences. As for tissues, the ViP signatures were trained on respiratory samples (nasal swabs, bronchoalveolar lavage fluid) and blood. However, in the original MS describing this signature (now published; Sahoo D et al., EBioMedicine, 2021), we presented evidence that the fundamental core host response (which is captured in the 166-gene ViP signature) was induced in diverse tissues, including liver, brain, heart, etc.

ACTION(S) TAKEN: We have expanded the **introduction section (Page #4)** and **Results section (Page #5)** to include pertinent details on how the ViP signature was derived and how it was trained. Some of these edits also address the questions below.

We have also provided a new Figure (**Figure 3B**) in which, as requested by Reviewer #1, we analyzed numerous datasets representative of viral and no-viral infectious, autoimmune, and immunosuppressed conditions datasets originating from diverse tissue samples (blood, PBMCs, isolated immune cells, bronchoalveolar lavage, skin) to demonstrate that the ViP signatures identify a core fundamental host response mechanism that is *invariably conserved across* tissue types and diverse disease states (i.e., appears to be *promiscuous*), and yet, is somewhat *specific* to immune response to infections. [See Page #5 of rebuttal]

Comment 4) Throughout the paper, the authors claimed that the 20-gene sViP signature outperformed the 166-gene ViP signature in predicting disease severity in a number of cohorts but it is unclear which classification method was employed and whether the classifier was retrained for each cohort.

RESPONSE- Comment 4: In the original MS describing this signature (now published; Sahoo D et al., EBioMedicine, 2021), we had reported the discovery of a set of ViP signatures that were trained on pandemics of the past (Influenza and avian flu), and used without further training to prospectively analyze the samples from the current pandemic (COVID-19). These signatures were used as is in this work, and not retrained in KD, MIS-C or any other datasets of autoimmune diseases that have now been added to this revised manuscript.

ACTION(S) TAKEN: We have expanded the **introduction section (Page #4)** and **Results section (Page #5)** to include pertinent details on how the ViP signature was derived and how it was trained. Some of these edits also address the questions below.

REFERENCES CITED IN REBUTTAL:

- 1 Du, H. *et al.* Clinical characteristics of 182 pediatric COVID-19 patients with different severities and allergic status. *Allergy* **76**, 510-532, doi:10.1111/all.14452 (2021).
- 2 Kim, Y. H. *et al.* Prognostic usefulness of eosinopenia in the pediatric intensive care unit. *J Korean Med Sci* **28**, 114-119, doi:10.3346/jkms.2013.28.1.114 (2013).
- 3 Liu, X. *et al.* Comparative analysis of clinical characteristics, imaging and laboratory findings of different age groups with COVID-19. *Indian J Med Microbiol* **38**, 87-93, doi:10.4103/ijmm.IJMM_20_133 (2020).
- 4 Zhang, J. J. *et al.* Clinical characteristics of 140 patients infected with SARS-CoV-2 in Wuhan, China. *Allergy* **75**, 1730-1741, doi:10.1111/all.14238 (2020).
- 5 Du, Y. *et al.* Clinical Features of 85 Fatal Cases of COVID-19 from Wuhan. A Retrospective Observational Study. *Am J Respir Crit Care Med* **201**, 1372-1379, doi:10.1164/rccm.202003-0543OC (2020).
- 6 Liu, F. *et al.* Patients of COVID-19 may benefit from sustained Lopinavir-combined regimen and the increase of Eosinophil may predict the outcome of COVID-19 progression. *Int J Infect Dis* **95**, 183-191, doi:10.1016/j.ijid.2020.03.013 (2020).
- 7 Roca, E., Ventura, L., Zattra, C. M. & Lombardi, C. EOSINOPENIA: an early, effective and relevant COVID-19 biomarker? *QJM: An International Journal of Medicine* **114**, 68-69, doi:10.1093/qjmed/hcaa259 (2020).
- 8 Andreozzi, F., Hermans, C. & Yombi, J. C. Eosinopenia and COVID-19 patients: So specific ? *EClinicalMedicine* **24**, 100439, doi:10.1016/j.eclinm.2020.100439 (2020).
- 9 Gruber, C. N. *et al.* Mapping Systemic Inflammation and Antibody Responses in Multisystem Inflammatory Syndrome in Children (MIS-C). *Cell* **183**, 982-995.e914, doi:10.1016/j.cell.2020.09.034 (2020).
- 10 Vella, L. A. *et al.* Deep immune profiling of MIS-C demonstrates marked but transient immune activation compared to adult and pediatric COVID-19. *Sci Immunol* **6**, doi:10.1126/sciimmunol.abf7570 (2021).
- 11 Carter, M. J. *et al.* Peripheral immunophenotypes in children with multisystem inflammatory syndrome associated with SARS-CoV-2 infection. *Nat Med* **26**, 1701-1707, doi:10.1038/s41591-020-1054-6 (2020).
- 12 Wright, V. J. *et al.* Diagnosis of Kawasaki Disease Using a Minimal Whole-Blood Gene Expression Signature. *JAMA Pediatr* **172**, e182293, doi:10.1001/jamapediatrics.2018.2293 (2018).

REVIEWER COMMENTS

Reviewer #1 (Remarks to the Author):

We thank the authors for arduously sorting through our numerous prior comments and suggestions, addressing these, and correcting our misinterpretation of the data from FigS1 in which a correlation is shown for KD13 in MISC but not KD. We especially commend the authors for exploring how VIP and sVIP gene sets are represented in non-infectious hyperinflammatory syndromes; we believe the inclusion of these additional analyses have further elevated the impact of their findings and their conclusions. Furthermore, their more comprehensive presentation of the statistics that were omitted in the prior version (e.g. the KD severity statistics) are very helpful in shedding insight on what is/isn't different between KD and MISC. Finally, the inclusion of the additional references and the correction of the arrangement of these citations are appreciated.

While we believe that the manuscript is significantly improved, a few minor issues still exist; we believe that these can readily be remedied. These are included below, in the order of their appearance in the manuscript:

- 1) Given that MISC is a post-infectious syndrome, the rationale for using a gene-set trained on acute viral infections should be further explained. This was requested previously, but we still do not feel that it has been adequately addressed in the revised manuscript. This is an important point to address, as one would expect that proximal immune responses would be most highly elevated at the time of acute infection, and perhaps not weeks after convalescence – this would be a discussion that would be of value to the reader.
- 2) In line 80, “various” should be changed to “variably”
- 3) In lines 118, 221, 225, 332, etc, what is the meaning of “invariant” host response? We believe this term could be better defined. In other words, while we are aware of innate immune responses and invariant cells of the immune response (e.g. iNKT cells), we are not sure what is meant by “invariant host response”. If the authors are using this term to imply a set of common proximal immune responses to a variety of respiratory viral infections, it may be beneficial to state this explicitly.
- 4) In Figure 4D through H, the ranked list metric portion of each graph appears with the same zero cross-over listed for each (22144). Could the authors please check this and correct as needed?
- 5) We are still not convinced that Figures 6F and G should be included in this manuscript, and it seems like authors wish to force a square peg through a round hole. This figure could certainly be the foundation of a separate case report, but in our opinion, the non-controlled n of 1 observation made here just does not appear to meet the standards of a Nature Communications data element; this opinion actually seems to be acknowledged by the authors in the Response to Reviewers, in which they state “While we acknowledge that n=1 and n=0 is not credible (beyond the purposes of case reports), that IL15 is induced in the heart is not the main message of the paper. Instead, it is our

attempt to connect hyperinflammation and one of the target organs.” The issue is that this figure is not showing evidence of hyperinflammation, but is instead indeed showing IL15 and IL15R expression, and these alone are not convincing as evidence of hyperinflammation, especially as IL15 appears to be a pro-survival factor in cardiomyocytes (PMID 24805144), suggesting that there may be a number of alternative explanations for these findings – explanations that may or may not be related to hyperinflammation.

Reviewer #2 (Remarks to the Author):

I must say that the authors have made substantial improvements of their work.

Reviewer #3 (Remarks to the Author):

The publication of the ViP and sViP signatures in EBioMedicine is very helpful. However, my comment #4 was not addressed at all in the revision. It is still unclear how the training and testing were done. As the 20-gene sViP signature and the 166-gene ViP signature were identified from all the cohorts studied, there was some information leakage when such signatures were used again to predict outcomes in the same set of cohorts. For each cohort studied here, the authors should leave it out for signature identification and training and then testing the performance for the signatures derived from the leave-one-out experiments.

My additional concern is the lack of a comparison with the standard approaches (e.g., intersection of ACE2 correlated DEG signatures in different diseases) for identifying signatures shared by multiple diseases as well as signatures specific to each disease. This analysis will give a more comprehensive picture about the similarity and difference between MIS-C and Kawasaki disease at the molecular level.

Reviewer #1:

General comment: We thank the authors for arduously sorting through our numerous prior comments and suggestions, addressing these, and correcting our misinterpretation of the data from FigS1 in which a correlation is shown for KD13 in MISC but not KD. We especially commend the authors for exploring how VIP and sVIP gene sets are represented in non-infectious hyperinflammatory syndromes; we believe the inclusion of these additional analyses have further elevated the impact of their findings and their conclusions. Furthermore, their more comprehensive presentation of the statistics that were omitted in the prior version (e.g. the KD severity statistics) are very helpful in shedding insight on what is/isn't different between KD and MISC. Finally, the inclusion of the additional references and the correction of the arrangement of these citations are appreciated.

General Response to Reviewer #1: We appreciate that the reviewer accurately summarized the diligence that went into revising the manuscript during round #1. We are glad to see that he/she is satisfied with those changes. We appreciate this generally positive feedback. During this second round of revision, we have addressed the few minor comments and critiques from this reviewer. We really appreciate the reviewer's attention to detail and time/effort that went into making these helpful suggestions, all of which we believe made the work more comprehensive.

Minor comments: While we believe that the manuscript is significantly improved, a few **minor issues** still exist; we believe that these can readily be remedied. These are included below, in the order of their appearance in the manuscript:

Comment 1) Given that MISC is a post-infectious syndrome, the rationale for using a gene-set trained on acute viral infections should be further explained. This was requested previously, but we still do not feel that it has been adequately addressed in the revised manuscript. This is an important point to address, as one would expect that proximal immune responses would be most highly elevated at the time of acute infection, and perhaps not weeks after convalescence – this would be a discussion that would be of value to the reader.

Response 1) We agree. We have now highlighted this rationale in both 'Introduction' (last paragraph on **Page 4**) and "Discussion" (on **Page #12**). We noted two recent studies that have revealed a prolonged presence of viral replication (PMID:34032635) and dendritic cell exhaustion caused by the persistence of the antigen (PMID:34599760).

Comment 2) In line 80, "various" should be changed to "variably"

Response 1) Correction has been made.

Comment 3) In lines 118, 221, 225, 332, etc, what is the meaning of "invariant" host response? We believe this term could be better defined. In other words, while we are aware of innate immune responses and invariant cells of the immune response (e.g. iNKT cells), we are not sure what is meant by "invariant host response". If the authors are using this term to imply a set of common proximal immune responses to a variety of respiratory viral infections, it may be beneficial to state this explicitly.

Response 3) We agree. The reviewer's interpretation (second one) is accurate. We have now explicitly defined it at the very beginning (when the word invariant is used first, and in its corresponding Figure 1 legend) and either removed the work or replaced the word later in the manuscript with 'shared'. Each edit has been **highlighted** in the revised manuscript.

Comment 4) In Figure 4D through H, the ranked list metric portion of each graph appears with the same zero cross-over listed for each (22144). Could the authors please check this and correct as needed?

Response 4) We have crosschecked our GSEA analyses. All the GSEA analyses were performed using gseapy pre-ranked algorithm where the ranking was obtained based on the differential expression patterns between Acute KD and MIS-C. Since the ranking was same in all the GSEA analyses, the zero-crossing number of the ranked list metric is also the same. Therefore, we expect same zero-crossing number in all the panels.

Comment 5) We are still not convinced that Figures 6F and G should be included in this manuscript, and it seems like authors wish to force a square peg through a round hole. This figure could certainly be the foundation of a separate case report, but in our opinion, the non-controlled n of 1 observation made here just does not appear to meet the standards of a Nature Communications data element; this opinion actually seems to be acknowledged by the authors in the Response to Reviewers, in which they state “While we acknowledge that n=1 and n=0 is not credible (beyond the purposes of case reports), that IL15 is induced in the heart is not the main message of the paper. Instead, it is our attempt to connect hyperinflammation and one of the target organs.” The issue is that this figure is not showing evidence of hyperinflammation, but is instead indeed showing IL15 and IL15R expression, and these alone are not convincing as evidence of hyperinflammation, especially as IL15 appears to be a pro-survival factor in cardiomyocytes (PMID 24805144), suggesting that there may be a number of alternative explanations for these findings – explanations that may or may not be related to hyperinflammation.

Response 5) We agreed with the reviewer is that n = 1 is not a rigorous kind of claim. Where we disagreed with the reviewer is that despite its case report worthy nature, we authors felt that the data are still valuable and add to an otherwise rigorous piece of work for a couple of reasons.

First, the ability to get heart tissue at autopsy in a disease as rarely fatal as KD is not trivial. We were fortunate to have this ability to ask and answer a simple question—i.e., is the cytokine upregulation associated with IL15 seen in the cardiac muscles? Being able to see/show that is the case at the tissue level supports the central role of this pathway in inflammation.

Second, we were also aware of the literature that IL15 appears to be a pro-survival factor in cardiomyocytes (PMID 24805144), and the mechanism claimed by which IL15 aids such survival was claimed to be surviving ER stress. Ongoing studies are squarely addressing the next chapter in this puzzle—i.e., does surviving ER stress due to IL15 expression contribute to senescence-associated secretory phenotype (SASP) and cardiac fibrosis (which is the major late complication in KD).

Thus, we continue to see it slightly differently from this reviewer’s opinion; while we acknowledge that this is case report worthy, we remain in favor of using this rare (and precious) tissue resource and evidence as a ‘cherry on the cake’ and not the cake itself.

Actions taken: Out of respect this reviewer’s opinion, and to address his/her concerns, we have explicitly cautioned the readers to not conclude too much about the panel 6F-G (new text added on **Page# 12**). If the acceptance of the manuscript is contingent upon us removing this figure, we will do so.

Reviewer #2:

I must say that the authors have made substantial improvements of their work.

Response to Reviewer #2: We are pleased to see that there are no remaining concerns and appreciate the time/effort that this reviewer put into reviewing this work.

Reviewer #3:

Comment 1: The publication of the ViP and sViP signatures in EBioMedicine is very helpful. However, my comment #4 was not addressed at all in the revision. It is still unclear how the training and testing were done. As the 20-gene sViP signature and the 166-gene ViP signature were identified from all the cohorts studied, there was some information leakage when such signatures were used again to predict outcomes in the same set of cohorts. For each cohort studied here, the authors should leave it out for signature identification and training and then testing the performance for the signatures derived from the leave-one-out experiments.

Response 1: We are grateful for this comment from reviewer#3, in which he/she points out some weaknesses in our description of the training and testing dataset in the previous work (published in the journal EBioMedicine) reporting the discovery of the ViP/sViP signatures. We understand that addressing these weaknesses is vital for the present work to be interpretable as a stand-alone work.

Actions taken: In this revised version, we have clarified the specifics about how the ViP signatures were derived. We inserted the information in the main text to reduce confusion (see **Page #4**). Two datasets (GSE47963, n = 438; GSE113211, n = 118) were used to *train* the ViP signature (**Fig 2** of the EBioMedicine paper), and sViP signature was obtained using one additional dataset (GSE101702, n = 159; Fig 4 of the EBioMedicine paper). We state explicitly that *KD/MIS-C datasets were not used during the training* of the ViP signature but tested prospectively in this current manuscript. Thus, information leakage is unlikely, and no additional leave-one-out experiments are needed.

The various places within the manuscript where we have now included this information include:

- 1) Main text:
 - a. Introduction (last paragraph on Page #4)
 - b. Results on Page #5.
 - c. Methods: ViP signatures
- 2) Supplementary Online Materials: Computational Methods, Subtitle- *ViP and severe (s)ViP Signatures*

Comment 2: My additional concern is the lack of a comparison with the standard approaches (e.g., intersection of ACE2 correlated DEG signatures in different diseases) for identifying signatures shared by multiple diseases as well as signatures specific to each disease. This analysis will give a more comprehensive picture about the similarity and difference between MIS-C and Kawasaki disease at the molecular level.

Response 2: We agree.

Actions taken: We have added comparison with standard approaches (PCA, Hierarchical Agglomerative Clustering, DEG, **Fig 3E-I**) as requested by the reviewer.

PCA (**Fig 3F**) and Hierarchical Agglomerative Clustering (**Fig 3G**) demonstrated two distinct clusters: one of them was enriched with MIS-C samples and other was enriched with KD samples. We removed three MIS-C (**Fig 3F**, yellow) and three KD (**Fig 3F**, cyan) samples that did not cluster with the other samples in the PC analysis and identified differentially expressed genes between seven KD (**Fig 3F**, green) and seven MIS-C (**Fig 3F**, orange) samples.

Reactome analysis of the genes up-regulated in MIS-C revealed interferon and cytokine signaling which were confirmed in the serum cytokine array (MSD, **Fig 4B**) and GSEA of the RNASeq (**Fig 4D-H**).

Venn diagram showed 11 genes overlapping (**Fig 3I**) between ViP signature and up-regulated genes in MIS-C. However, no genes were overlapped between ViP signature and down-regulated genes in MIS-C. These analyses strengthened our hypothesis that KD and MIS-C share molecular features, with MIS-C in the extreme end of that spectrum.

Results on **Page #10** and Discussion section on **Pages 13** reflect these additions. A new excel file (**Supplemental Information 3**) has been uploaded, which contains the list of DEGs that resulted from the hierarchical agglomerative clustering. We hope that these additions mitigate any remaining concerns of this reviewer.

REVIEWERS' COMMENTS

Reviewer #1 (Remarks to the Author):

Thank you for allowing me to once again participate in the review of this outstanding and illuminating manuscript. I thank the authors for acknowledging that my suggestions and comments have strengthened the manuscript. In the second revision of this manuscript, Dr. Ghosh and colleagues have addressed all of my previous concerns, except for those regarding the inclusion of an n of 1 data element and the interpretations that ensue from this (Figures 6F and G). At this point however, I do not wish to further delay the publication of this otherwise excellent manuscript, and will leave the final decision as to how to proceed with the editor.

Reviewer #3 (Remarks to the Author):

My concerns were well addressed. I have no more concern.

ORIGINAL COMMENTS

REVIEWERS' COMMENTS

Reviewer #1 (Remarks to the Author):

Thank you for allowing me to once again participate in the review of this outstanding and illuminating manuscript. I thank the authors for acknowledging that my suggestions and comments have strengthened the manuscript. In the second revision of this manuscript, Dr. Ghosh and colleagues have addressed all of my previous concerns, except for those regarding the inclusion of an n of 1 data element and the interpretations that ensue from this (Figures 6F and G). At this point however, I do not wish to further delay the publication of this otherwise excellent manuscript, and will leave the final decision as to how to proceed with the editor.

Reviewer #3 (Remarks to the Author):

My concerns were well addressed. I have no more concern.

RESPONSE TO REVIEWER'S COMMENTS

We are pleased to see that for the most part both reviewers were satisfied with the revisions that we made during the last submission. Reviewer #1 has explicitly stated this, using generous words of praise (e.g., outstanding, illuminating, and excellent), which is very encouraging. We continue to acknowledge that low fatality in KD and MIS-C makes it impossible to obtain cardiac tissue at autopsy, which limits our ability to go beyond proof of concept (n = 1) study in Fig 6. We have explicitly stated that in various places within the manuscript, including figure legend, text and in the section on 'study limitations'.